# POC-SLT: Partial Object Completion with SDF Latent Transformers

## Abstract

3D geometric shape completion hinges on representation learning and a deep understanding of geometric data. Without profound insights into the three-dimensional nature of the data, this task remains unattainable. Our work addresses this challenge of 3D shape completion given partial observations by proposing a transformer operating on the latent space representing Signed Distance Fields (SDFs). Instead of a monolithic volume, the SDF of an object is partitioned into smaller high-resolution patches leading to a sequence of latent codes. The approach relies on a smooth latent space encoding learned via a variational autoencoder (VAE), trained on millions of 3D patches. We employ an efficient masked autoencoder transformer to complete partial sequences into comprehensive shapes in latent space. Our approach is extensively evaluated on partial observations from ShapeNet and the ABC dataset where only fractions of the objects are given. The proposed POC-SLT architecture compares favorably with several baseline state-of-the-art methods, demonstrating a significant improvement in 3D shape completion, both qualitatively and quantitatively.

## 1 Introduction

The 3D measurements from scanners or sensors can be used to assemble virtual objects or entire worlds. This can be used not only for visualizations but also for reasoning about the world, e.g., in self-driving cars. In general, with optical sensors, it is only possible to measure 3D surfaces that are visible to the sensor from within its range of motion (if any). Even after combining all observations taken from different angles, gaps remain in the observed surface. One can usually not observe an object from all directions, e.g., from below the surface it is standing on. Similarly, one might only be able to take pictures from the front of an object but may not be able to peek behind it.

In this work, we present a new method for partial object completion using a transformer in the style of Masked Auto Encoders (MAE) (He et al., 2022) operating in latent SDF space. For a unknown or masked regions, it is supposed to fill in a plausible shape that connects well with the given geometry. The method consumes and produces shapes represented as Signed Distance Fields (SDF) on volumetric patches. Similar to Stable Diffusion (Rombach et al., 2022), the incomplete shapes are first encoded into a smooth, compressed latent space representation using our dedicated *Patch Variational Autoencoder (P-VAE)*. For high-resolution models, the entire shape is represented as a sequence of latent codes generated by the P-VAE on volumetric patches. Completion is performed in latent space and the completed shape is decoded back to an SDF using the decoder of the P-VAE on each patch. Note that our *SDF-Latent-Transformer* is a Masked Autoencoder. It therefore solves the task directly in a single inference step without a transformer decoder.

Shape completion requires a strong prior for real-world shapes in order to estimate the most likely shapes for incomplete inputs. We trained our P-VAE and SDF-Latent-Transformer on ShapeNet-CoreV1 (Chang et al., 2015) with all 55 categories. We further fine-tuned the transformer on a subset of the ABC dataset (Koch et al., 2019) with 100K meshes.

By thoroughly evaluating our methods on shape completion tasks, we demonstrate superior quality compared to state of the art. Our key contributions to the field are: i) A comprehensive Patch Variational Autoencoder (P-VAE) to compress SDF shapes into sequences of codes in a smooth latent space. ii) A SDF-Latent-Transformer trained as Masked Autoencoder completes input shapes within milliseconds in a single inference step. iii) Upon acceptance, we will release code and model

checkpoints and make the training data sets available. For the P-VAE, it consists of 2.6 million $32^3$ patches and for the SDF-Latent-Transformer of $128^3$ SDFs for all objects in ShapeNetCoreV1.

Ours is the first SDF-based method trained and evaluated on all of ShapeNet ($\approx$50k meshes) and a 100k subset of ABC. This is an order of magnitude more than previous SDF-based approaches, at higher resolution, while using consumer-grade GPUs. MAEs are used in many domains by now but are usually a means to pretrain a powerful encoder, whose strong representation can be used for downstream tasks. We however directly use the training objective as a shortcut to get instant completion. This only works due to the robustness and strong geometric prior in our latent space.

## 2 RELATED WORK

We review related point-based, occupancy-based, and SDF-based 3D completion methods and methods inspiring our deep-learning architecture.

**Point-based Shape Completion.**   3D Shape completion tackles the task of reconstructing 3D objects from partial observations in the form of 2D images, sometimes with depth information, or 3D point clouds. Areas occluded during capture need to be filled in to form a complete object.

PointNet and its variations (Qi et al., 2017a;b) have pioneered the direct processing of 3D coordinates with neural networks, catalyzing research in numerous downstream tasks, including point cloud completion, as for example PCN (Yuan et al., 2018). PCN uses PointNet in an encoder-decoder framework and integrates a FoldingNet (Yang et al., 2018) to transpose 2D points onto a 3D surface by simulating the deformation of a 2D plane. Following PCN, a plethora of other methods have emerged (Tchapmi et al., 2019; Huang et al., 2020; Xie et al., 2020; Liu et al., 2019), aiming to enhance point cloud completion with higher resolution and increased robustness.

PointTR (Yu et al., 2021) was one of the first approaches which used transformers (Vaswani et al., 2017) for point cloud completion. They derive the point proxies from a fixed count of furthest point sampled representatives, which are turned into a feature vector using a DGCNN (Wang et al., 2019). Completion is done by a geometry-aware transformer, which is queried for missing point proxies. The missing points are extracted from the predicted proxies using a FoldingNet (Yang et al., 2018). The completed points are finally merged with the input point cloud. AdaPointTR (Yu et al., 2023) addresses discontinuity issues and is more robust to noisy input. SeedFormer (Zhou et al., 2022) completes a sparse set of patch seeds from an incomplete input and upsamples the seeds to a complete point cloud. The seeds are sparse 3D positions enriched by a transformer with semantic information about the local neighborhood. SnowFlakeNet (Xiang et al., 2021) solves this decoding task with a SkipTransformer-based architecture, which repeatedly splits and refines low-resolution points. LAKe-Net (Tang et al., 2022) first localizes aligned keypoints to generate a surface skeleton mesh which aids in producing a complete point cloud. AnchorFormer (Chen et al., 2023) extracts global features and predicts key "anchor" points which are combined with a subset of input points and then upsampled into a dense mesh. VRPCN (Pan et al., 2021) models the shape from the partial input as probability distributions, which are sampled and then refined by a hierarchical encoder-decoder network. This is related to our approach as we also use a Variational Auto Encoder (VAE) to compress the shape in latent space. LION (Zeng et al., 2022) introduced a flexible latent diffusion model for point clouds also using a VAE to map point cloud features into latent space. The features and diffusion model are constructed from Point-Voxel CNNs.

**Occupancy-based Shape Completion.**   Occupancy Networks (Mescheder et al., 2019) introduced a functional, implicit representation of 3D shapes via their occupancy. These methods can be queried for arbitrary 3D locations and will return whether the queried position is inside or outside of the represented object. Meshes can quickly be recovered from this representation using the MISE algorithm (Mescheder et al., 2019). Peng et al. (2020) extend this to a convolutional method and represent entire scenes by storing and interpolating latent representations in 2D/3D grids. Chibane et al. (2020) combine local and global information using multi-scale feature grids in IF-Nets.

These methods focus on identifying and representing 3D objects seen in images and adding detail to uniformly low-resolution point cloud and voxel input.

ShapeFormer (Yan et al., 2022) compactly encodes shapes into a sparse grid of vector-quantized deep implicit functions (VQDIF). Partial inputs are completed by an autoregressive transformer operating on a sequence of location and patch index pairs. 3DILG (Zhang et al., 2022) uses an irregular grid of latent codes for shape representation and completes partial inputs with an autoregressive transformer. 3DShape2VecSet (Zhang et al., 2023) derives multiple global vectors for shape representation using cross-attention between input points and sampled query points. The (variationally) auto-encoded shape is then completed using latent-space diffusion. The output occupancy is queried by cross-attention between a dense point cloud and the global vectors.

The input to these methods is typically a point cloud generated from a single camera view. Occluded parts need to be inferred from the partial point cloud. In many cases, the input still covers large parts of the object, exposing global information about the shape, while our method completes objects from a small subset of complete patches.

**SDF-based Shape Completion.** SDFs represent 3D shapes implicitly as continuous 3D scalar functions which describe the distance of any 3D point to the closest surface. The surface is given by the zero-level-set while in- or outside areas are identified by the sign. This ensures that objects represented by SDFs are always watertight. In contrast, meshes are often non-manifold since they are commonly created to "look right" on screen, rather than to represent the volume of solid geometry. While SDFs could be modeled in any functional basis, we use regular 3D grids.

There are only few approaches related to shape completion on SDFs in the literature. For SDF representation and class-related shape generation, DeepSDF (Park et al., 2019) proposes to use an AutoDecoder (Tan & Mavrovouniotis, 1995) to jointly optimize a compressed latent representation of an SDF and the decoder to extract the full SDF from the latent representations. Due to the fully connected layer, the approach has limited resolution. Our approach instead splits the SDF into patches, which are encoded with a rigorously trained VAE (Kingma & Welling, 2013). Those patches can be assembled to very highly detailed shapes. We experimented with using a similar AutoDecoder optimization scheme to refine latent codes for our frozen decoder but found that the optimized latent codes complicate subsequent shape completion tasks (Appendix A.1).

To represent entire scenes, LIG (Jiang et al., 2020) autoencodes TSDF patches and operates on oriented points during test time. They divide the scene into multi-scale patch-grids and use trilinear interpolation to produce seamless results. Similar to their work, we divide objects into patches and also found medium patch-sizes to be most effective: Patches need to be small enough to generalize across objects but large enough to represent interesting geometric features.

AutoSDF and SDFusion (Mittal et al., 2022; Cheng et al., 2023) focus on multi-modal shape generation and completion in latent space, e.g. from an input image, point clouds, or depth maps. Both approaches use Vector Quantized Variational Auto Encoders (VQ-VAEs) (van den Oord et al., 2017; Razavi et al., 2019) to patchwise compress SDF shapes to a compact latent space. SDFusion then uses a latent diffusion model, while AutoSDF employs an autoregressive transformer to generate or complete shapes sequentially. AutoSDF (Mittal et al., 2022) is in many regards similar to our approach. The main differences are: i) the choice of a VAE over a VQ-VAE to avoid quantization errors. ii) Larger patches with $32^3$ grid points over the $8^3$ grid points used by AutoSDF. iii) Our transformer is trained as Masked Auto Encoder (MAE) (He et al., 2022) while the AutoSDF Transformer is trained autoregressively. This drastically changes how the model can be used as our approach does not require a sequential generation. iv) Our approach is trained on SDF grids with a resolution of $128^3$ and trained on the full ShapeNetCoreV1 (Chang et al., 2015) and refined on the ABC datasets (Koch et al., 2019) instead of the SDF resolution of $64^3$ on a 13-category subset of SapeNetCoreV1 for AutoSDF.

PatchComplete (Rao et al., 2022) and DiffComplete (Chu et al., 2023) both complete $32^3$ Truncated SDF input using multi-scale features. While PatchComplete uses an attention mechanism, DiffComplete uses diffusion and also allows combination of multiple inputs. Both methods learn strong priors that generalize well to new inputs, albeit at comparably low resolution. While our method works on a subset of clean SDF patches to extend partial geometry, these methods work on the 3D-EPN (Dai et al., 2017) dataset, containing noisy, low-resolution partial SDFs generated from a single camera view.

**Variational Autoencoders (VAE)** (Kingma & Welling, 2013) are networks that learn latent representations of input data. Unlike standard autoencoders, they learn a probability distribution over the latent space. This generates a smooth latent space which allows manipulation and computations in latent space. Inspired by Stable Diffusion (Rombach et al., 2022), we train a VAE to encode the $32^3$ SDF patches into a smooth latent space. We call this network Patch Variational Autoencoder (P-VAE). A full $128^3$ SDF grid is therefore represented by only 64 latent tokens. This makes any further processing very efficient and scalable.

**Transformers** (Vaswani et al., 2017) are a network architecture for sequence-to-sequence tasks. It uses attention as the core mechanism to identify pairwise relationships between elements in a sequence. They have revolutionized Natural Language Processing and are the foundation of all current Large Language Models. Transformers can however process any sequence of data. In Dosovitskiy et al. (2020), images are split into a sequence of patches, which is then processed by transformers. The approach in He et al. (2022) masks out some of those patches and train to fill in the masked-out patches. Similarly, we split the SDF grid into patches and encode them into a sequence of latent codes. We mask out parts of the latent codes that contain unknown geometry and use our SDF-Latent-Transformer to complete the missing patches.

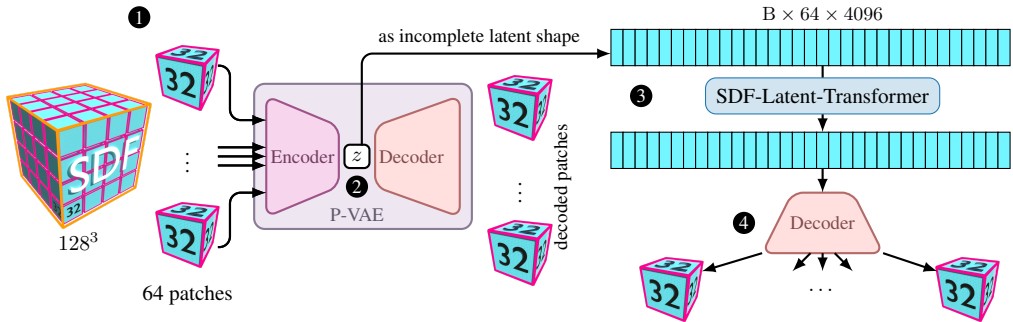

Figure 1: Architecture Overview. The $128^3$ input SDF is divided into $4 \times 4 \times 4$ patches of size $32^3$. For each patch, a latent code is generated by a variational autoencoder (P-VAE) resulting in the input stream to our SDF-Latent-Transformer, which can be masked to produce partial inputs. It is trained as a Masked AutoEncoder (MAE) to generate a completed series of tokens which are finally translated back to SDF patches using the P-VAE decoder.

## 3 METHOD

Our POC-SLT shape completion pipeline consists of four main steps which are depicted in Figure 1: First, the input $128^3$ SDF grid is split into $4 \times 4 \times 4$ patches of size $32^3$ ❶. Each of those patches is encoded into a latent vector using the P-VAE ❷. The latent codes $z$ that form the input shape are then assembled in a sequence, which is masked and processed by the SDF-Latent-Transformer (SLT) ❸. Finally, the resulting sequence is decoded patch-by-patch with the decoder from the P-VAE into a completed SDF ❹. We will now elaborate on these four steps.

### 3.1 P-VAE

The task for the P-VAE is to encode $32^3$-SDF grid patches $(p_i)_{i=0,...,n}$ into latent code vectors $(z_i)_{i=0,...,n}$. For details about the architecture and implementation of our P-VAE, refer to Appendix D. The P-VAE consists of an encoder $E_{\mathrm{VAE}}$ and a decoder $D_{\mathrm{VAE}}$, which are both based on 3D-convolutional layers. A patch $p_i$ is encoded by the encoder into mean $\mu_i$ and variance $\sigma_i^2$ of a Gaussian distribution. During training, this distribution is sampled to obtain the latent code $z_i \sim \mathcal{N}(\mu_i, \sigma_i^2)$, which is then decoded by the Decoder $D_{\mathrm{VAE}}$ into an SDF patch $\tilde{p}_i = D_{\mathrm{VAE}}(z_i)$. We use the mean absolute error between $p_i$ and $\tilde{p}_i$ as reconstruction loss. To regularize the latent space of the VAE, we follow the *KL-regularized* VAE implementation from Stable Diffusion (Rombach et al., 2022). During inference, we directly use the means $z_i = \mu_i$ without additional sampling.

The P-VAE is trained on SDF patches extracted randomly from parts of the meshes of ShapeNet-CoreV1 (Chang et al., 2015). Each patch has a fixed SDF resolution of $32^3$. We however strongly

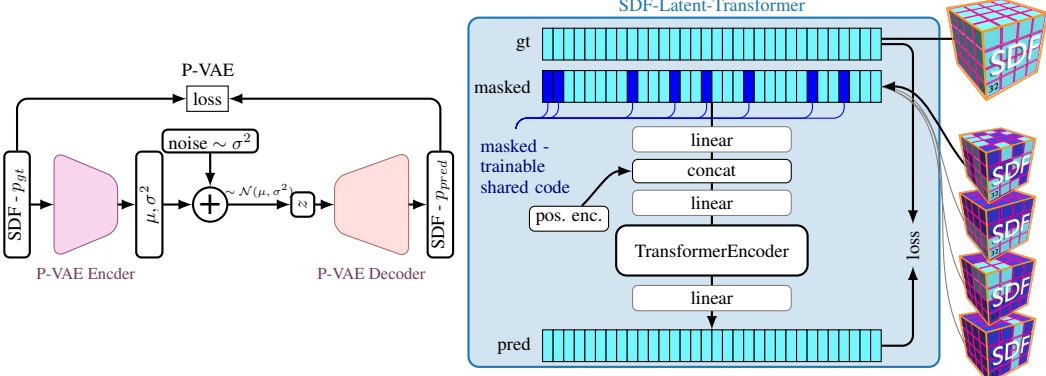

Figure 2: **P-VAE (left)**: On millions of patches, we train a smooth embedding space for SDFs with a variational autoencoder (P-VAE) that samples the noise according to the predicted variance before passing the estimated latent to the decoder. **SDF-Latent-Transformer (SLT) (right)**: Performs shape completion on input sequences consisting of SDF-Patches (cyan) in latent space. During training, some of the input patches are masked and substituted with a trainable shared vector (blue). The utilized masking schemes *Random*, *Half*, *Octant* and *Slice* are visualized on the right. 3D positional encoding is added before a TransformerEncoder propagates the information from the remaining patches to all the masked patches completing the 3D shape.

vary the side length of each extracted patch while ensuring that they remain close to a surface. Specifically, we uniformly sample surface points and side lengths $d \sim \mathcal{U}_{[0.1,1]}$ as well as offsets $x, y, z \sim \mathcal{N}(0, d/3)$. The P-VAE is therefore exposed to a large variety of detail levels, scales, and surface types and has to learn the full variety of how patches of natural shapes can potentially look like. This yields a strong, generalizable shape prior which allows the P-VAE to generalize even to patches of shapes from completely different datasets.

## 3.2 SDF-LATENT-TRANSFORMER

The actual shape completion happens completely in latent space. Our approach is visualized on the right in Figure 2. The incomplete input shape is provided as a sequence of latent space vectors. We employ a standard transformer encoder as a Masked Auto Encoder to fill in missing latent vectors. The missing input parts are marked by a special trainable *mask token* (blue in the figure). The positions of all vectors, including masked ones, are provided to the transformer as 3D positional encoding (Sitzmann et al., 2020). Linear layers before and after concatenating the positional embedding have empirically shown to improve convergence. Finally, the transformer completes the sequence.

**Outside SDF Patches.** For many, especially anisotropic shapes, some of the encoded SDF-Patches will not contain any geometry either because they are completely inside or outside an object. Note however, that those patches still contain valid distance values towards surfaces that lie outside those patches. We therefore do not differentiate between patches that explicitly contain surfaces and those that lie outside.

**Masking during Training.** The training strategy of the SDF-Latent-Transformer is visualized in Figure 2. For creating the mask for a given training example, we randomly choose from one of the following masking strategies: 1. *Random Masking*, 2. *Octant*, 3. *Half*, 4. *Slice*. With *Random Masking*, each patch is masked out with a $40\%$ probability. With *Octant*, all but one of the octants of the patch-grid are masked out. With *Half*, one half of the patch-grid is masked out. With *Slice*, everything but one axis-aligned slice through the patch-grid is masked out. The four different masking schemes are visualized in the same order from top to bottom on the right side of Figure 2. We use *Random Masking* with a probability of $35\%$ and all the other strategies with a probability of $21.7\%$ each.

**Loss Functions.** We supervise our training with ground truth patch encodings $z = E_{\text{VAE}}$ for the complete object. The entire training of the SDF-Latent-Transformer happens in latent space. The full loss function $\mathcal{L}_{\text{comp}}$ consists of two terms:

$$\mathcal{L}_{\text{comp}} = \alpha \mathcal{L}_{\text{masked}} + \beta \mathcal{L}_{\text{non-masked}} \tag{1}$$

Both $\mathcal{L}_{\text{masked}}$ and $\mathcal{L}_{\text{non-masked}}$ are simple $L1$-losses on the latent codes evaluated on results with masked and non-masked inputs respectively. The weights $\alpha$ and $\beta$ are chosen, such that the contribution of every patch to the total loss is equal, no matter how many patches were masked. With $N$ patches in total from which $M$ are masked out we get:

$$\alpha = \frac{M}{N} \qquad\qquad \beta = \frac{N - M}{N} \tag{2}$$

**Shape Completion.** For shape completion, one can use the SDF-Latent-Transformer in the same mode as during training. We assume the partial shape is given in the form of a high-resolution SDF volume. For each given patch, a latent code is generated with the P-VAE encoder. The known SDF patches (including known empty patches) are handed over as input to the transformer while marking all others as masked. The transformer will then predict proper latent codes for all masked tokens, which includes predicting latent codes for empty patches.

The resulting grid of latent codes can then be decoded back into an $128^3$ SDF which can be converted into a mesh, e.g. using Marching Cubes (Lorensen & Cline, 1987).

## 4 EVALUATION

POC-SLT is an efficient and fast solution for patch-wise SDF completion in latent space. It consists of the P-VAE which encodes SDF patches into latent space and the SDF-Latent-Transformer which fills in missing patches in latent space. To demonstrate the effectiveness of our method, we measure its performance on various completion tasks and compare it to related work below. Additional tasks are demonstrated in Appendix A. All metrics used in the experiments are defined in Appendix B. Data preparation for turning meshes into SDFs is described in Appendix C.

**Completion.** First, we provide quantitative and qualitative results for three SDF shape completion tasks on the full ShapeNetCoreV1 (Chang et al., 2015) dataset and ABC (Koch et al., 2019) test sets. The tasks are to complete an SDF based on different types of partial inputs. 1. Only the bottom half of the SDF is given as input (Half). 2. Only the bottom right octant is given as input (Oct). 3. Patches are removed randomly with 25%, 50%, and 75% of the SDF remaining (R25, R50, R75).

A simpler version of those tasks on a subset of ShapeNet has been suggested by Wu et al. (2020) and used by (Wu et al., 2020; Yu et al., 2021; Mittal et al., 2022; Cheng et al., 2023) for completion comparisons. Here, only 13 categories of ShapeNet are used for training and evaluation.

In addition, we also compare to AnchorFormer (Chen et al., 2023) on the (Half) task, by using point cloud inputs with only points in the bottom half of the bounding box (Half). Note that this does not represent a typical input for their method.

**P-VAE.** The P-VAE is evaluated by measuring the deviation between the input of the encoder and the output of the decoder. We furthermore compare the reconstructions from our P-VAE latent space to the reconstructions of an expensively optimized latent space using AutoDecoding (Park et al., 2019; Tan & Mavrovouniotis, 1995). We also compare against the VQDIF by Yan et al. (2022).

**Timing.** Our latent space completion consists of a single forward step of the SLT. This makes the shape completion inference very fast. We demonstrate this advantage over previous latent shape completion works (Mittal et al., 2022; Yan et al., 2022; Cheng et al., 2023) in Table 1.

**Hardware.** The P-VAE and SDF-Latent-Transformer variations were trained on a machine with 3 Nvidia RTX 4090 GPUs and 512GB of main memory. The P-VAE was trained for approximately two weeks, while the transformers were trained for about two days each. Additional computation was required for preprocessing (Appendix C) and evaluation, which was carried out over several machines in our cluster, using mostly Nvidia RTX 2080 Ti GPUs and several CPUs.

Table 1: Comparing shape completion inference time, resolution, and number of model parameters (encode/decode + completion).

| Method | Inference Time | Output Resolution | Num Params |
|---|---|---|---|
| AutoSDF (Mittal et al., 2022) | 4.3 seconds | $64^3$ | 33.3 M + 34.0 M |
| ShapeFormer (Yan et al., 2022) | 20 seconds | $128^3$ | 23 M + 317 M |
| SDFusion (Cheng et al., 2023) | 6.6 seconds | $64^3$ | 26.4 M + 620.7 M |
| POC-SLT (Ours) | 8.6 milliseconds | $128^3$ | 60.3 M + 503.0 M |

## 4.1 COMPLETION

The SDF-Latent-Transformer is the core component of our completion pipeline. It receives an incomplete sequence of SDF patches in latent space and completes it to a full sequence of latent codes in a single forward step. The latent codes of the completed sequence are independently decoded into SDF patches with the P-VAE decoder. The patches are trivially assembled into a high-resolution shape volume from which we extract surface meshes using Marching Cubes (Lorensen & Cline, 1987). We trained two slightly different variants of the SDF-Latent-Transformer: **SLT:** trained on ShapeNetCoreV1 with latent codes $z$ from P-VAE as ground truth and **SLT ABC:** SLT fine-tuned on a subset of the ABC dataset. The quality of our shape completion approach is evaluated on unseen objects from ShapeNetCoreV1 (Chang et al., 2015) and ABC (Koch et al., 2019).

**Qualitative Results (ShapeNet).**   Examples for the previously described bottom half (Half) completion task on ShapeNetCoreV1 (Chang et al., 2015) are shown in Figure 3. Here, we compare our SLT against AutoSDF (Mittal et al., 2022) and AnchorFormer (Chen et al., 2023) as recent state-of-the-art methods.

Our POC-SLT pipeline generates highly plausible shapes. Both AutoSDF (Mittal et al., 2022) and AnchorFormer (Chen et al., 2023) struggle to fill in the missing parts. Compared to AutoSDF, our method works at a significantly higher resolution and can handle all ShapeNet classes. The higher resolution is for example important for the fine structures on the bench table in column four or the faucet in column two. AutoSDF (Mittal et al., 2022) completely fails to reconstruct the car in column one or the armchair in column three, while our SLT on the same input produces correct results. AnchorFormer (Chen et al., 2023) often produces sparse and simplistic completions for inputs with unknown top halves which is especially visible for the faucet in column two. However, the sparseness and therefore lack of detail are also visible in the car in the first column, the armchair in the second, and the cupboard in the last column.

**Qualitative Results (ABC).**   Shape completion results from halves (Half) or octants (Oct) on ABC (Koch et al., 2019) are visualized in Figure 4. For this task, we use SLT ABC, which was fine-tuned on the ABC dataset. The ABC dataset contains many planar and rotationally symmetric objects. These symmetries are picked up by the SDF-Latent-Transformer to complete missing parts with high detail and high plausibility, such as the object in column one or column four.

On the other hand, the provided input does not always constrain the output sufficiently. This can lead to deviations when compared to ground truth while still generating plausible objects, such as the upper spokes in the third column, the unsymmetric object in the sixth column, or the missing hole in the last column of the figure.

**Generalization (Objaverse).**   To demonstrate the generalizability of our approach we evaluate the SLT, trained only on ShapeNet (Chang et al., 2015), on the (Half) task with objects from Objaverse (Deitke et al., 2023). The type of objects shown in Figure 5 have never been seen by the SLT. As expected, the results show clear deviations from the ground truth, especially for objects which are far away from the kind of objects that are found in ShapeNet, such as the animals in column four and five. For simpler objects, such as the pillow in column three or the mushroom in column one, our approach produces good completions. There is a chair category in ShapeNet, which allows our model to plausibly complete the rocking chair in column two.

**Quantitative Results.**   We numerically evaluate the previously described tasks in Table 2.

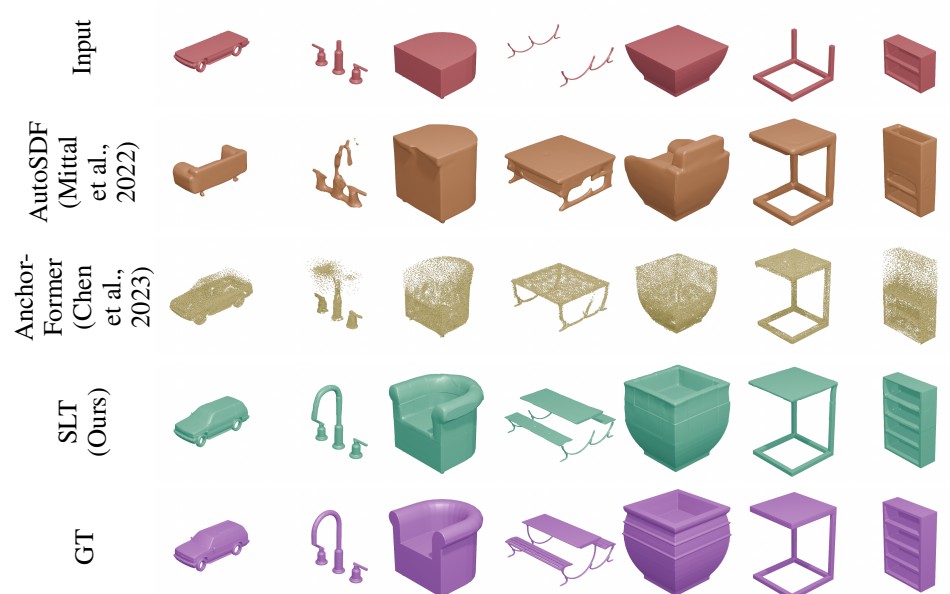

Figure 3: Completion of ShapeNet (Chang et al., 2015) objects from bottom half. Comparison to AutoSDF (Mittal et al., 2022) and AnchorFormer Chen et al. (2023). Our SLT completes these objects more plausibly than AutoSDF. The density of completed points by AnchorFromer drastically varies in the completed regions.

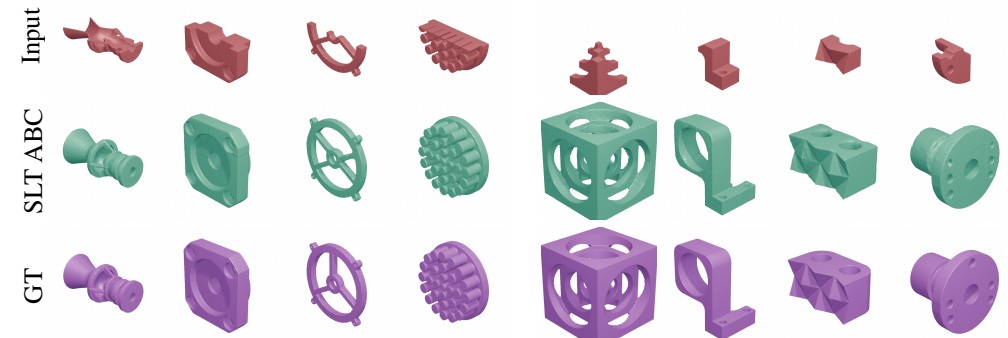

Figure 4: Completion of ABC (Koch et al., 2019) objects from the bottom half (left) and octant (right). The SLT learned to complete partially symmetric objects quite successfully.

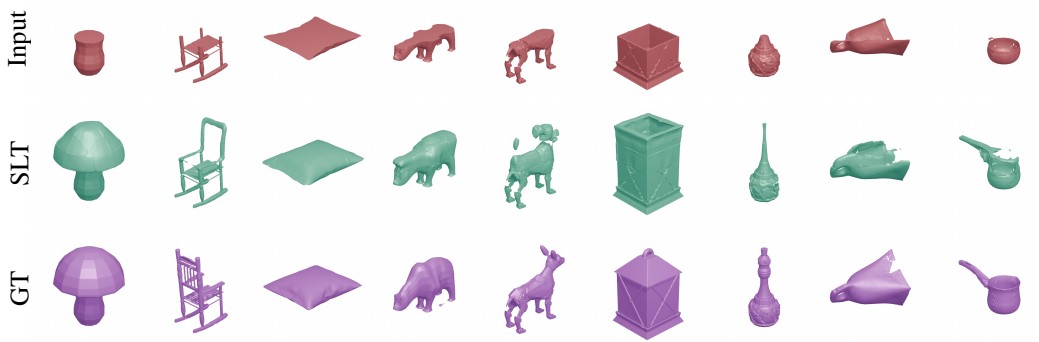

Figure 5: Completion of out-of-distribution objects from Objaverse (Deitke et al., 2023) using the SLT trained on ShapeNet (Chang et al., 2015). The last three columns show completions of scanned real-world objects.

Table 2: Evaluation on shape completion tasks (Half), (Oct), (R75), (R50) and (R25) with the ShapeNetCoreV1 (Chang et al., 2015) dataset with all 55 categories and on (Half) and (Oct) with the ABC (Koch et al., 2019) dataset. Details on the metrics can be found in Appendix B.

| Model | Dataset | Task | IoU↑ | $F_1$↑ | CD↓ | HD↓ | NC↑ | IN↓ | CMP↑ |
|---|---|---|---|---|---|---|---|---|---|
| SLT | SN | Half | 0.7466 | 0.8468 | 1.0221 | 0.0765 | 0.9200 | 0.4196 | 0.9067 |
| SLT | SN | Oct | 0.5884 | 0.7336 | 1.2467 | 0.0966 | 0.8589 | 0.6034 | 0.8404 |
| SLT | SN | R75 | 0.9153 | 0.9792 | 0.2258 | 0.0452 | 0.9677 | 0.2862 | 0.9905 |
| SLT | SN | R50 | 0.8650 | 0.9495 | 0.4829 | 0.0595 | 0.9504 | 0.3512 | 0.9751 |
| SLT | SN | R25 | 0.7645 | 0.8677 | 0.8567 | 0.0789 | 0.9183 | 0.4512 | 0.9329 |
| SLT ABC | ABC | Half | 0.8617 | 0.9159 | 0.8703 | 0.0575 | 0.9435 | 0.2551 | 0.9466 |
| SLT ABC | ABC | Oct | 0.7144 | 0.7744 | 2.9247 | 0.1077 | 0.8779 | 0.3986 | 0.8391 |

**Comparison.** We compare with previous works designed for the (Half) task on the 13 category ShapeNetCoreV1 subset defined by Wu et al. (2020). The results in Table 3 show that our method significantly outperforms AutoSDF (Mittal et al., 2022) and SDFusion (Cheng et al., 2023).

Table 3: Shape completion from (Half) on all categories of the ShapeNetCoreV1 subset defined by Wu et al. (2020). * Please note that we compare voxel-based IoU. While we evaluate our method at $128^3$, other methods only produce $64^3$ results, where it is much easier to achieve a high IoU.

| Method | UHD↓ | IoU*↑ | $F_1$↑ | CD↓ | HD↓ | NC↑ | IN↓ | CMP↑ |
|---|---|---|---|---|---|---|---|---|
| AutoSDF (Mittal et al., 2022) | 0.0618 | **0.9824** | *0.6785* | *6.1828* | *0.1871* | *0.7698* | *0.7381* | *0.7970* |
| SDFusion (Cheng et al., 2023) | *0.0548* | *0.9728* | 0.6170 | 13.3824 | 0.2590 | 0.7272 | 0.8091 | 0.6170 |
| SLT (ours, retrained on subset) | **0.0354** | 0.7263 | **0.8295** | **3.3519** | **0.0937** | **0.9079** | **0.4469** | **0.8924** |

## 4.2 P-VAE & AUTODECODER

The P-VAE accepts SDF patches of size $32^3$ as input, encodes them into compressed latent representations and decodes latent representations back to SDF patches. It was trained and evaluated on a training split of ShapeNet. We evaluate the P-VAE on unseen data in Table 4 and compare our results with the VQDIF by Chang et al. (2015). Our numeric evaluation demonstrates the high quality of our latent space. In comparison the VQDIF, which only operates on 32k input points produces coarse reconstructions, e.g. Figure 6. Note that the P-VAE was not fine-tuned for the following experiments on ABC (Koch et al., 2019) or Objaverse (Deitke et al., 2023), demonstrating the strong generalizability resulting from our training scheme.

Table 4: Encode-decode performance on ShapeNet (Chang et al., 2015). We compare our P-VAE against VQDIF by ShapeFormer (Yan et al., 2022). Note that the VQDIF by ShapeFormer operates on 32k points per object as input whereas our method operates on $128^3$ SDFs per object.

| Method | IoU↑ | $F_1$↑ | CD↓ | HD↓ |
|---|---|---|---|---|
| VQDIF (Yan et al., 2022) | 0.6026 | 0.6552 | 56.4902 | 0.2170 |
| P-VAE (ours) | **0.9482** | **0.9931** | **0.0001** | **0.0264** |

Park et al. (2019) suggested to use an AutoDecoder (AD) (Tan & Mavrovouniotis, 1995) to optimize the latent codes for a known SDF such that the frozen decoder $D_{\text{VAE}}$ produces the best possible result. The optimized latent codes should yield more accurate surface reconstruction than the codes directly produced by the encoder $E_{\text{VAE}}$ in a single forward pass. However, the time spent on refinement is orders of magnitude longer, e.g. around $0.04$ sec for the P-VAE encoder vs. 25 sec for the AutoDecoder per object.

We compare our P-VAE to the *AutoDecoder* approach by encoding a given SDF patch $p_{gt}$ to a latent code $z = E_{\text{VAE}}(p_{gt})$. We then add random noise to $z$ to get an initial version $z'$, which is then

decoded into a SDF $p_{pred} = D_{\text{VAE}}(z')$. We compute a loss $|p_{gt} - p_{pred}|$ to update $z'$ in a loop till convergence.

Both P-VAE and AD produce high-quality results with only minor differences to ground truth. Across the dataset, we get a tiny improvement over the Hausdorff Distance from 0.026 to 0.024 from the optimization. These improvements are mostly visible for objects with fine detail as shown in Figure 6. One could use these optimized latent codes $z'$ to train the SLT towards producing such output tokens at no extra cost. We experimented on this idea and show results in Appendix A.1.

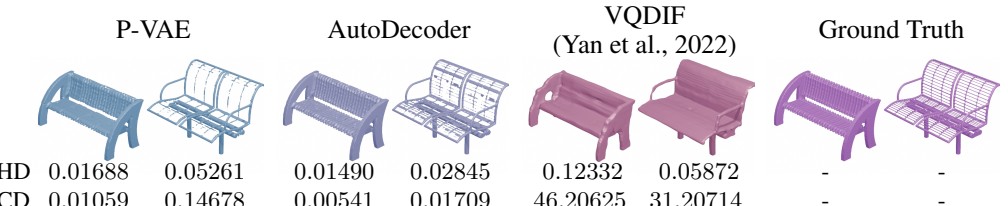

| | P-VAE | | AutoDecoder | | VQDIF (Yan et al., 2022) | | Ground Truth | |
|---|---|---|---|---|---|---|---|---|
| HD | 0.01688 | 0.05261 | 0.01490 | 0.02845 | 0.12332 | 0.05872 | - | - |
| CD | 0.01059 | 0.14678 | 0.00541 | 0.01709 | 46.20625 | 31.20714 | - | - |

Figure 6: Comparison of latent codes generated by P-VAE encoding and refined via the AutoDecoder. While overall subtle, these extreme examples demonstrate the benefit of using the AutoDecoder to refine the patch embeddings. Our P-VAE represents significantly higher levels of detail than the Vector-Quantized Deep Implicit Functions (VQDIF) by Yan et al. (2022).

## 5 LIMITATIONS

While the presented results are of high quality, some downstream tasks for the POC-SLT pipeline might need additional work. At the moment, it is trained assuming a fixed bounding box. This is a valid assumption for most objects but might be too limiting for open scenes. In principle, the transformer should be able to cope with sequences of basically arbitrary length. However, it still needs to be investigated if the current model can deal with higher-resolution spatial encodings to make use of additional tokens.

The completion is performed for masked patches. Implicitly, the approach always assumes either completely given SDF patches or completely unknown patches. In completion applications for images, depth maps, or partial 3D scans, one might additionally need to indicate that the patch information itself might be incomplete, e.g. a missing occluded surface in the same cell.

We do not consider the single-view 3D reconstruction task. Previous work (Tatarchenko et al., 2019) has shown that predicting object-centered results heavily relies on identifying (and augmenting) similar objects in the training set and can even be outperformed by object retrieval. Lacking a single global object representation, our method is not well-suited for such a classification task. Instead, one could first estimate a canonical pose, monocular depth, and then compute 3D SDF patches from the then recovered partial geometry in order to meaningfully perform geometric 3D reconstruction with our method. Evaluation of such an approach would highly depend on the models employed for pose estimation and depth estimation in addition to our model, preventing meaningful comparisons.

## 6 CONCLUSION

With POC-SLT, we proposed an accurate and efficient new method for SDF shape completion in latent space. POC-SLT processes SDFs in patches of fixed size. Two main components are used to refine and fill in missing patches in a shape. Firstly, an extensively trained Patch Variational Autoencoder (P-VAE) for accurately compressing the patches into a sequence of latent codes and back. Secondly, an SDF-Latent-Transformer (SLT) which completes and refines the latent sequence of an incomplete shape in a single inference step. We demonstrate that our approach produces highly accurate and plausible 3D shape completions, outperforming prior works. POC-SLT is trained on ShapeNetCoreV1 (Chang et al., 2015), is class agnostic, and can easily be adapted to new datasets, which we demonstrated with the ABC (Koch et al., 2019) dataset. We show that the P-VAE works across different datasets, even without additional training, and hope that it will be a helpful tool for future research. Extending the approach to deal with partial information within patches and developing applications like real-time scan completions are left open for future work.

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

# A ADDITIONAL EXPERIMENTS

In this section we show several additional experiments which did not fit into the main paper. We evaluate training the SLT on auto-decoded latent codes (Section A.1), inspect how the SLT trained on ShapeNet (Chang et al., 2015) can generalize to the ABC dataset (Section A.2), show visual results of shape completion on the completion tasks (R25), (R50) and (R75) with randomly masked inputs (Section A.3), compare our latent space representation with DeepSDF (Park et al., 2019) and 3DShape2VecSet (Zhang et al., 2023) (Section A.4), provide evaluation on 3D-EPN Dai et al. (2017) (Section A.6), and, finally, present further comparisons with AnchorFormer (Chen et al., 2023) in Section A.7.

## A.1 SDF-LATENT-TRANSFORMER ON REFINED LATENT CODES

The comparison in Figure 6 suggests that the latent codes for some objects might be improved by explicitly optimizing the latent codes of the patches using the AutoDecoder technique (Park et al., 2019; Tan & Mavrovouniotis, 1995). To test if the SLT might improve performance when trained on regular partial latent codes $z$ as input but with optimized $z'$ as ground truth, we optimized the latent codes for all objects in the training dataset of ShapeNet (Chang et al., 2015). We trained a separate SDF Latent Transformer on this dataset called SLT-AD, expecting the SLT-AD to also learn

to do the expensive AutoDecoder optimization for free in its forward pass. We also tested to run the regular SLT and, on the resulting SDF, the SLT-AD without masking just for refinement. This model is called SLT + SLT-AD. Finally, we tried running the SLT-AD twice: Once on the incomplete input and then again on the completed output, which is called SLT-AD + SLT-AD.

The quantitative results are shown in Table 5 and qualitative results are shown in Figure 7. The numbers for the bottom half (Half) experiment clearly suggest, that the expected improvement did not happen. The results for the octant (Oct) experiment are inconclusive at best.

While the AutoDecoder technique (Park et al., 2019; Tan & Mavrovouniotis, 1995) produces latent codes that decode to a more accurate representation of the given input, we believe that the latent codes generated this way are more likely to be outliers and thus not as easily understood and utilized by the SLT-AD, resulting in worse performance.

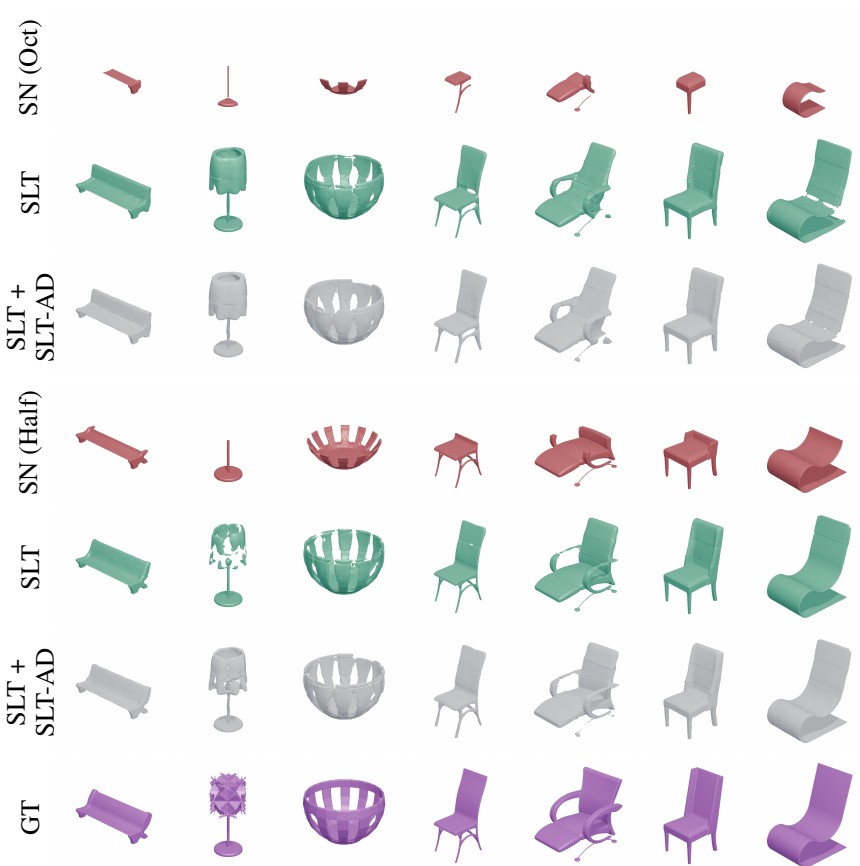

Figure 7: Completion from halves and octants by our SLT and then refined by SLT-AD on ShapeNet.

### A.2    ABC COMPLETION USING SLT TRAINED ON SHAPENET

To test the generalizability of our SLT trained on ShapeNet (Chang et al., 2015), we evaluate it on the ABC (Koch et al., 2019) dataset without fine-tuning. We report the metrics for the tasks (Half) and (Oct) in Table 6. To compare to the fine-tuned SLT ABC, we duplicate its numbers reported in Table 2 here for convenience. As expected, the reconstruction metrics of the fine-tuned model are much better. Figure 8 shows, however, that the STL still does a reasonably good job at completing many of the objects from ABC (Koch et al., 2019). Instead of producing complete mechanical parts, it often creates chairs, tables, and vases from their partial inputs and it picks up surprisingly well on some of the inherent symmetries. With the (Oct) input, the SLT has more freedom to complete the input into objects found in ShapeNet (Chang et al., 2015), such as chairs, e.g., column four.

Table 5: Shape completion on ShapeNet (Chang et al., 2015) with several versions of the SLT. SLT is the default version, SLT-AD hs been trained on auto-decoded ground-truth latent codes, and the "+ SLT-AD" variants feed the previous output through SLT-AD again without masking for refinement.

| Model | Task | IoU↑ | $F_1$↑ | CD↓ | HD↓ | NC↑ | IN↓ | CMP↑ |
|---|---|---|---|---|---|---|---|---|
| SLT | Half | **0.7466** | **0.8468** | 1.0221 | **0.0765** | **0.9200** | **0.4196** | 0.9067 |
| SLT-AD | Half | 0.7237 | 0.8344 | 1.2065 | 0.0846 | 0.9119 | 0.4767 | **0.9086** |
| SLT + SLT-AD | Half | 0.6867 | 0.8101 | 1.0039 | 0.0811 | 0.8976 | 0.5268 | 0.8696 |
| SLT-AD + SLT-AD | Half | 0.6780 | 0.7969 | **0.9111** | 0.0880 | 0.8923 | 0.5528 | 0.8658 |
| SLT | Oct | 0.5884 | **0.7336** | **1.2467** | 0.0966 | 0.8589 | 0.6034 | **0.8404** |
| SLT-AD | Oct | **0.6127** | 0.7121 | 1.8994 | 0.1010 | **0.8694** | **0.5704** | 0.8279 |
| SLT + SLT-AD | Oct | 0.5878 | 0.6848 | 1.6883 | 0.0970 | 0.8557 | 0.6052 | 0.7729 |
| SLT-AD + SLT-AD | Oct | 0.5912 | 0.6795 | 1.4225 | **0.0940** | 0.8560 | 0.6167 | 0.7767 |

Table 6: Evaluation on completion tasks (Half), (Oct) on the ABC (Koch et al., 2019) dataset using SLT and comparing to the SLT ABC results copied from Table 2. The results report the mean over all categories. Details on the metrics can be found in Appendix B.

| Model | Dataset | Task | IoU↑ | $F_1$↑ | CD↓ | HD↓ | NC↑ | IN↓ | CMP↑ |
|---|---|---|---|---|---|---|---|---|---|
| SLT | ABC | Half | 0.7478 | 0.8123 | 2.0658 | 0.1053 | 0.9134 | 0.3730 | 0.8869 |
| SLT ABC | ABC | Half | **0.8617** | **0.9159** | **0.8703** | **0.0575** | **0.9435** | **0.2551** | **0.9466** |
| SLT | ABC | Oct | 0.5710 | 0.6304 | 5.1872 | 0.1532 | 0.8208 | 0.5433 | 0.7234 |
| SLT ABC | ABC | Oct | **0.7144** | **0.7744** | **2.9247** | **0.1077** | **0.8779** | **0.3986** | **0.8391** |

### A.3 COMPLETION ON RANDOMLY MASKED SHAPENET

One of the masking strategies during training is to randomly mask out inputs as outlined in Section 3.2. We also numerically evaluated the random masking completion tasks (R75), (R50) and (R25) in Table 2. Now, in Figure 9 we show visual results on ShapeNet (Chang et al., 2015), for the same three completion tasks. Even when only small parts of the input are given, e.g. 25% in the (R25) task, the information in neighboring tokens is used to produce near-perfect completions. This is in contrast to the significantly more challenging (Half) and (Oct) tasks we chose to evaluate our method where completion needs to happen most often for non-neighbor patches.

### A.4 LATENT QUALITY COMPARISON WITH DEEPSDF AND 3DSHAPE2VECSET

We compare the auto-encoding quality of our P-VAE (Section 3.1) against DeepSDF (Park et al., 2019) in Table 7 and against 3DShape2VecSet (Zhang et al., 2023) in Table 8. For 3DShape2VecSet (Zhang et al., 2023), we follow their evaluation code with regards to the size of the object and with regards to the L1 Chamfer Distance. Like in their code, we omit the division by 2 which would be part of the original definition from the supplemental of Mescheder et al. (2019). In addition to the numbers on our regular P-VAE, we also report the results when decoding optimized latent codes $z'$ using an AutoDecoder as described in Section A.1 under the label P-VAE-AD.

The results in Table 7 demonstrate that, on autoencoding quality, our P-VAE outperforms DeepSDF on all categories with and without optimized latent codes. The numbers in Table 8 suggest that the autoencoding performance of our P-VAE is significantly better in $F_1$ and $CD_{L1}$ and on par in IoU with the cross-attention based autoencoding strategy from 3DShape2VecSet.

### A.5 SHAPE COMPLETION ON SHAPENET

We present shape completion results on the ShapeNetCoreV1 subset defined by Wu et al. (2020), compatible with numbers reported by prior methods, in Table 9. This subset uses only 13 categories of ShapeNet for training and evaluation. In contrast to Table 3, here, only the *chair* category is considered, and only UHD is reported, since those are the only compatible values reported by previous

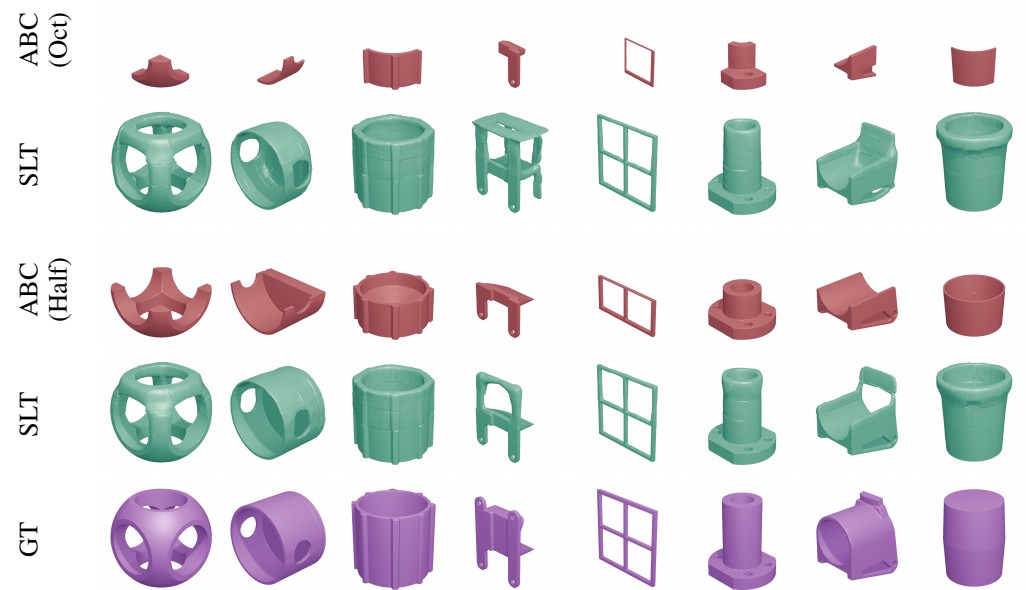

Figure 8: Testing generalizability via completion of ABC (Koch et al., 2019) objects using the SLT trained exclusively on ShapeNet (Chang et al., 2015).

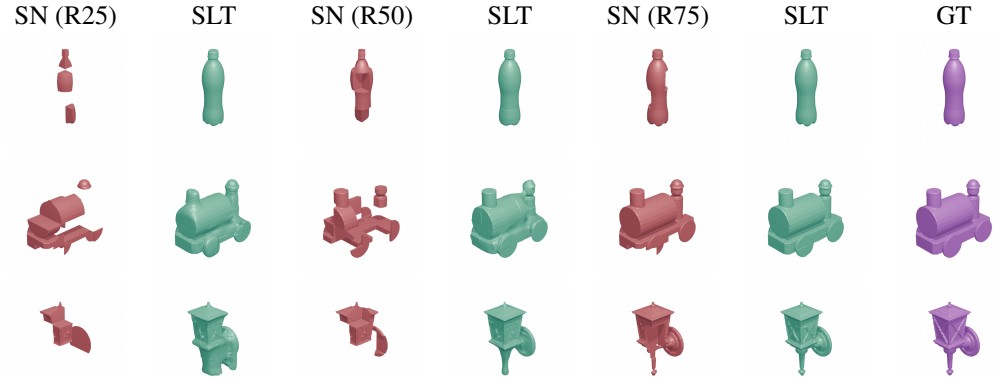

Figure 9: Completion of randomly masked ShapeNet (Chang et al., 2015) objects using the SLT. This is often a slightly simpler task as small holes are relatively easily filled by propagating information from neighboring patches.

work. Note that this only measures how faithfully the geometry given in the input was preserved during the completion.

## A.6    SHAPE COMPLETION ON 3D-EPN

While our model was never intended for and never trained to perform shape completion from incomplete SDF patches, it might still be interesting to see how it performs on such a task. Incomplete SDF patches mean that, within the same patch, parts of the geometry are already missing.

In Figure 10, we show the result of trying to complete the partial $32^3$ SDFs in the 3D-EPN (Dai et al., 2017) dataset with our SLT trained on complete, high-resolution ShapeNet (Chang et al., 2015) patches. The partial SDFs from 3D-EPN (Dai et al., 2017) are generated from a single camera view and, thus, everything behind the first visible surface is considered inside, often producing incorrect SDF patches where outside areas are encoded to be deep inside an object. We upsample the low

Table 7: Comparing mean and median (L2) Chamfer Distance in a P-VAE auto-encoding quality comparison with DeepSDF (Park et al., 2019).

| CD↓ (mean) | chair | plane | table | lamp | sofa |
|---|---|---|---|---|---|
| P-VAE | **0.0759** | 0.0236 | 0.0914 | 0.415 | 0.0269 |
| P-VAE-AD | 0.1427 | **0.0177** | **0.0743** | **0.4041** | **0.0263** |
| DeepSDF | 0.204 | 0.143 | 0.553 | 0.832 | 0.132 |
| CD↓ (median) | chair | plane | table | lamp | sofa |
| P-VAE | 0.0203 | 0.0075 | 0.023 | 0.0167 | 0.0264 |
| P-VAE-AD | **0.0199** | **0.0067** | **0.0225** | **0.0138** | **0.0259** |
| DeepSDF | 0.072 | 0.036 | 0.068 | 0.219 | 0.088 |

Table 8: P-VAE autoencoding quality comparison with 3DShape2VecSet (Zhang et al., 2023).

| Method | IoU↑ | $F_1$↑ | $CD_{L1}$ ↓ | HD↓ |
|---|---|---|---|---|
| P-VAE | 0.9483 | **0.9931** | **0.0094** | 0.0658 |
| 3DShape2VecSet | **0.965** | 0.970 | 0.038 | - |

resolution SDFs to our $128^3$ resolution in order to get multiple patches. This leads to staircase artifacts which have not been seen in this way by the SLT, even for complete patches.

Our SLT assumes that every given, non-masked $32^3$ input patch contains complete information. Therefore, it does not attempt to repair their partially corrupt information and – for the most part – just feeds the given input through to the output. While some masked patches on the side of the objects can be filled in, as expected, this generally leads to poor results.

While it could be possible to train or fine-tune the existing SLT architecture on this task, we think that further research would be required to best adapt our method to this setting.

### A.7 COMPARISON WITH ANCHORFORMER

Since our method works on SDF patches rather than point clouds, performing one-to-one comparisons with point cloud methods is somewhat problematic.

In Table 10, we show the performance of AnchorFormer (Chen et al., 2023) when evaluated on the regular PCN (Yuan et al., 2018) test split and when presented with bottom-half inputs from the same objects, uniformly sampled from our preprocessed (Appendix C) version of ShapeNet (Chang et al., 2015). We removed some of the objects from the PCN (Yuan et al., 2018) test split, mostly cars, for which we did not have a matching mesh available to generate bottom-half inputs.

For evaluating AnchorFormer (Chen et al., 2023), we use their only published pre-trained checkpoint which was trained on PCN (Yuan et al., 2018). Completion from just the bottom half can be a much harder task, since the output on the top half is completely unconstrained. Typically, PCN (Yuan

Table 9: Quantitative comparison on shape completion on the *chair* category of the ShapeNetCoreV1-subset with 13 categories from Wu et al. (2020). Completion tasks are (Half) and (Oct). In compliance with previous work, we report the Unidirectional Hausdorff Distance from input to completed result.

| Method/UHD↓ | Half | Oct |
|---|---|---|
| MPC (Wu et al., 2020) | 0.0627 | 0.0579 |
| PoinTR (Yu et al., 2021) | 0.0572 | 0.0536 |
| AutoSDF (Mittal et al., 2022) | 0.0567 | 0.0599 |
| SDFusion (Cheng et al., 2023) | 0.0557 | - |
| SLT (ours) | **0.0445** | **0.0467** |

Figure 10: Completion of partial $32^3$ SDFs from 3D-EPN (Dai et al., 2017) using our SLT which is trained only on complete patches from $128^3$ SDFs from ShapeNet (Chang et al., 2015). While not trained on this kind of input, the SLT still demonstrates its geometric understanding and produces reasonable shape completions.

et al., 2018) and other partial point cloud settings provide at least some sparse global information which guides and constrains the completion process.

As a second experiment (also shown in Table 10), we use the same AnchorFormer (Chen et al., 2023) model and checkpoint to complete our test set of ShapeNet (Chang et al., 2015) meshes from bottom halves, as shown in Figure 3 and evaluated for our method in Table 2.

The numbers on the PCN (Yuan et al., 2018) data show that AnchorFormer (Chen et al., 2023) is not well suited to solve the (Half) completion task. Both metrics drop significantly. Furthermore, on our ShapeNet (Chang et al., 2015) data, the (Half) task produces slightly better numbers than on the PCN (Yuan et al., 2018) data. However, our approach achieves better completion quality on the (Half) shape completion task.

Table 10: Comparison with AnchorFormer (Chen et al., 2023) on completing partial inputs from ShapeNet (Chang et al., 2015). Details on the compared configurations are explained in Section A.7. Some qualitative results can be seen as part of Figure 3.

| Method | Data Source | Split | Task | $F_1 \uparrow$ | $CD_{(L2)} \downarrow$ |
|---|---|---|---|---|---|
| AnchorFormer | PCN | PCN | PCN | 0.8379 | 0.1986 |
| AnchorFormer | ShapeNet | PCN | Half | 0.6953 | 1.4791 |
| AnchorFormer | ShapeNet | Ours | Half | 0.7164 | 4.2816 |
| Ours (see Table 2) | ShapeNet | Ours | Half | **0.8468** | **1.0221** |

## B  METRICS

In order to evaluate any of the following metrics, we use Marching Cubes (Lorensen & Cline, 1987) to generate a mesh from our generated SDF and then uniformly sample 1M points $X$ and $Y$ from the resulting meshes.

We use the following point-based metrics: Hausdorff Distance (HD), to measure the largest gap between the original and the reconstructed geometry:

$$\text{HD}(X, Y) = \max \left\{ \max_{x \in X} \left\{ \min_{y \in Y} d(x, y) \right\}, \max_{y \in Y} \left\{ \min_{x \in X} d(x, y) \right\} \right\}. \tag{3}$$

Similarly, the Unidirectional Hausdorff Distance (UHD) measures the maximum distance between objects, but here, in line with previous work (Mittal et al., 2022), it is measured from partial input $X$ to completion $Y$ to measure the "fidelity" of the completed output:

$$\text{UHD}(X, Y) = \max_{x \in X} \left\{ \min_{y \in Y} d(x, y) \right\}. \tag{4}$$

We use the ($L2$) Chamfer Distance (Fan et al., 2017), multiplied by 1000, to measure the accuracy of the reconstruction via the mean squared distance between the original and reconstructed geometry:

$$\text{CD}(X,Y) = \frac{1}{|X|} \sum_{x \in X} \min_{y \in Y} \left\{ d^2(x,y) \right\} + \frac{1}{|Y|} \sum_{y \in Y} \min_{x \in X} \left\{ d^2(y,x) \right\}. \tag{5}$$

In both cases, $d(x,y)$ measures the $L2$ distance between two points $x, y \in \mathbb{R}^3$.

For consistency with prior work, these metrics are evaluated at a normalized scale where the bounding box diameter is 1. This corresponds to the normalization of the ShapeNet (Chang et al., 2015) dataset.

Following the evaluation of AutoSDF (Mittal et al., 2022), we separately measure objectionable reconstruction artifacts using F-score @1% ($F_1$), normal consistency (NC) and inaccurate normals (IN). The F-score @1% ($F_1$) sets a threshold at 1% of the side length of the reconstructed volume (Tatarchenko et al., 2019), within which a neighboring point on the ground truth mesh needs to be found from a reconstruction sample for computing precision and vice-versa for recall. From these, the $F_1$ score is computed as usual.

Completeness (CMP) measures the recall within a threshold of $1.5\%$ of the side length, based on the implementation of Wu et al. (2020).

Normal Consistency (NC) measures the mean absolute dot-product between normals on the reconstructed surface and the closest ground-truth surface point:

$$NC(X,Y) = \frac{1}{|X|} \sum_{x \in X} \left\{ |n_x \cdot n_y| \ : \ y = \arg\min_{y \in Y} d(x,y) \right\} \tag{6}$$

The absolute value is taken to allow for flipped normals in the ground truth data. For the normals $n_x$ and $n_y$, we use the geometric face normals of the faces that generated the sampled points $x$ and $y$. Inaccurate Normals (IN) complements the NC measure by counting the percentage of normals which are outside of a 5-degree threshold of the normal of the closest ground-truth point.

We also measure Intersection over Union (IoU) based on the sign of the $128^3$ SDFs.

## C  DATA PREPARATION

In order to compute SDFs from meshes, we translate them such that their bounding box is centered at the origin and then uniformly scale them into $[-1, 1]^3$. Then, for ShapeNetCoreV1, we remeshed the mesh in Blender via voxelization using a voxel size of $0.008$. This was necessary to create manifold meshes from which SDFs could be computed. Admittedly, this resulted in the loss of some geometry which does not have any volumetric counterpart. Nevertheless, we consider this remeshed version our ground truth data for SDF-based shape completion training. For evaluation, we excluded objects which lost all geometry in the conversion. The meshes in the ABC dataset are of much higher quality. Here, we only normalized the meshes to $[-1, 1]^3$ and then densely sampled signed distances in a regular grid of size $128^3$.

## D  ARCHITECTURE DETAILS

We implemented all our models in PyTorch (Paszke et al., 2019) and trained using PyTorch Lightning (William Falcon, 2019).

The P-VAE is a variational autoencoder built with 3D convolutions. The encoder converts an SDF-patch of shape $[B, 1, 32, 32, 32]$ into a mean $\mu$ and variance $\sigma^2$ vector of shape $[B, 8192]$. To be precise, the encoder returns $\log(\sigma^2)$, from which we compute $\sigma^2$. During training, the latent representation of the SDF-patch is sampled from the normal distribution $z \sim \mathcal{N}(\mu, \sigma^2)$. During inference we use $z = \mu$. The architectures of the encoder and the decoder are given in Table 13 and Table 14 respectively. The ConvBlock specified in Table 11 and the DecoderLayer in Table 12 are reoccurring architectural structures in both the encoder and the decoder. Note that each layer specifies its predecessor in the *parent* column. Skip connections are joined by adding the output of two parents together.

Table 11: Architecture of a **ConvBlock** element as used in the P-VAE Encoder in Table 13

| Layer Name | Parent Name | Layer Parameters | Output Shape |
|---|---|---|---|
| Conv3D | - | kernel 3, stride 1, pad 1 | $[B, c_{out}, d, d, d]$ |
| BatchNorm3d | Conv3D | - | $[B, c_{out}, d, d, d]$ |
| ReLU | BatchNorm3d | - | $[B, c_{out}, d, d, d]$ |

Table 12: Architecture of a **DecoderLayer** as used in Table 14. If not otherwise noted, all layers use a kernel size of 3 ($[3, 3, 3]$), stride 1 and padding 1 ($[1, 1, 1]$). The input has shape $[B, c_{in}, d, d, d]$.

| Layer Name | Parent Name | Layer Parameters | Output Shape |
|---|---|---|---|
| Conv3D | layer input | - | $[B, c_{out}, d, d, d]$ |
| Conv3DTrans | layer input | stride 2, output padding 2 | $[B, c_{out}, 2d, 2d, 2d]$ |
| BatchNorm3D | Conv3D | - | $[B, c_{out}, d, d, d]$ |
| Upsample | BatchNorm3D | factor 2, trilinear | $[B, c_{out}, 2d, 2d, 2d]$ |
| Add | Conv3DTrans, Upsample | - | $[B, c_{out}, 2d, 2d, 2d]$ |
| ReLU | Add | - | $[B, c_{out}, 2d, 2d, 2d]$ |

# E   TRAINING DETAILS

## E.1   PATCH-VARIATIONAL AUTOENCODER

We trained the P-VAE for 250 epochs with early stopping at epoch 193 with initial learning rate of 1e-4 and CosineAnnealing scheduler. The batch size for the training was 128 per GPU.

## E.2   AUTODECODER

The refined codes $z'$ were optimized with Adam for 200 steps per patch. The gaussian noise that was added on the initial $z$ from the P-VAE was sampled from a gaussian distribution $\mathcal{N}(0, 1e^{-2})$.

## E.3   SDF LATENT TRANSFORMER

The masking ratio used for training is $0.4$. Our learning rate was $1e^{-5}$ and we use a cosine scheduler with 1400 warm-up steps for the transformer and trained for 120k steps. The batch size was 64 per GPU. All SLT configurations used the same training configuration and setup.

Table 13: Architecture of the **P-VAE Encoder**. If not otherwise noted, all layers use a kernel size of 3 ($[3, 3, 3]$) stride 1 and padding 1 ($[1, 1, 1]$). The input patch has shape $[B, 1, 32, 32, 32]$.

| Layer Name | Parent Name | Layer Parameters | Output Shape |
|---|---|---|---|
| ConvBlock-1 | SDF patch | - | $[B, 32, 32, 32, 32]$ |
| ConvBlock-2 | ConvBlock-1 | - | $[B, 32, 32, 32, 32]$ |
| ConvBlock-3 | ConvBlock-2 | - | $[B, 32, 32, 32, 32]$ |
| ConvBlock-4 | ConvBlock-3 | - | $[B, 32, 32, 32, 32]$ |
| Add-1 | ConvBlock-1, SDF patch | broadcasting | $[B, 32, 32, 32, 32]$ |
| MaxPool-1 | Add-1 | stride 2 | $[B, 32, 16, 16, 16]$ |
| Conv3D-1 | Add-1 | stride 2 | $[B, 32, 16, 16, 16]$ |
| Add-2 | MaxPool-1, Conv3D-1 | - | $[B, 32, 16, 16, 16]$ |
| Conv3D-2 | Add-2 | kernel 1, padding 0 | $[B, 32, 16, 16, 16]$ |
| ConvBlock-5 | Add-2 | - | $[B, 64, 16, 16, 16]$ |
| ConvBlock-6 | ConvBlock-5 | - | $[B, 64, 16, 16, 16]$ |
| ConvBlock-7 | ConvBlock-6 | - | $[B, 64, 16, 16, 16]$ |
| ConvBlock-8 | ConvBlock-7 | - | $[B, 64, 16, 16, 16]$ |
| Add-3 | Conv3D-1, ConvBlock-7 | - | $[B, 64, 16, 16, 16]$ |
| MaxPool-2 | Add-3 | stride 2 | $[B, 64, 8, 8, 8]$ |
| Conv3D-3 | Add-3 | stride 2 | $[B, 64, 8, 8, 8]$ |
| Add-4 | MaxPool-2, Conv3D-3 | - | $[B, 64, 8, 8, 8]$ |
| Conv3D-4 | Add-4 | kernel 1, padding 0 | $[B, 128, 8, 8, 8]$ |
| ConvBlock-9 | Add-4 | - | $[B, 128, 8, 8, 8]$ |
| ConvBlock-10 | ConvBlock-9 | - | $[B, 128, 8, 8, 8]$ |
| ConvBlock-11 | ConvBlock-10 | - | $[B, 128, 8, 8, 8]$ |
| ConvBlock-12 | ConvBlock-11 | - | $[B, 128, 8, 8, 8]$ |
| Add-5 | Conv3D-4, ConvBlock-12 | - | $[B, 128, 8, 8, 8]$ |
| MaxPool-3 | Add-5 | stride 2 | $[B, 128, 4, 4, 4]$ |
| Conv3D-5 | Add-5 | stride 2 | $[B, 128, 4, 4, 4]$ |
| Add-6 | MaxPool-3, Conv3D-5 | - | $[B, 128, 4, 4, 4]$ |
| Conv3D-6 | Add-4 | kernel 1, padding 0 | $[B, 256, 4, 4, 4]$ |
| ConvBlock-13 | Add-6 | - | $[B, 256, 4, 4, 4]$ |
| ConvBlock-14 | ConvBlock-13 | - | $[B, 256, 4, 4, 4]$ |
| ConvBlock-15 | ConvBlock-14 | - | $[B, 256, 4, 4, 4]$ |
| ConvBlock-16 | ConvBlock-15 | - | $[B, 256, 4, 4, 4]$ |
| Add-7 | Conv3D-4, ConvBlock-16 | - | $[B, 128, 8, 8, 8]$ |
| MaxPool-4 | Add-7 | stride 2 | $[B, 128, 2, 2, 2]$ |
| Conv3D-7 | Add-7 | stride 2 | $[B, 128, 2, 2, 2]$ |
| Add-8 | MaxPool-4, Conv3D-7 | - | $[B, 128, 2, 2, 2]$ |
| Conv3D-8 | Add-8 | kernel 1, padding 0 | $[B, 512, 2, 2, 2]$ |
| ConvBlock-17 | Add-8 | - | $[B, 512, 2, 2, 2]$ |
| ConvBlock-18 | ConvBlock-17 | - | $[B, 512, 2, 2, 2]$ |
| ConvBlock-19 | ConvBlock-18 | - | $[B, 512, 2, 2, 2]$ |
| ConvBlock-20 | ConvBlock-19 | - | $[B, 512, 2, 2, 2]$ |
| Add-9 | Conv3D-8, ConvBlock-20 | - | $[B, 512, 2, 2, 2]$ |
| Conv3D-9 | Add-9 | kernel 1, padding 0 | $[B, 1024, 2, 2, 2]$ |

Table 14: Architecture of the **P-VAE Decoder**. The decoder converts a input latent code with shape $[B, 8192] \rightarrow [B, 512, 2, 2, 2]$ into an SDF-patch of shape $[B, 1, 32, 32, 32]$.

| Layer Name | Parent Name | Layer Parameters | Output Shape |
|---|---|---|---|
| DecoderLayer-1 | latent code | $c_{in} = 512, c_{out} = 512$ | $[B, 512, 4, 4, 4]$ |
| DecoderLayer-2 | DecoderLayer-1 | $c_{in} = 512, c_{out} = 256$ | $[B, 256, 8, 8, 8]$ |
| DecoderLayer-3 | DecoderLayer-2 | $c_{in} = 256, c_{out} = 256$ | $[B, 256, 16, 16, 16]$ |
| DecoderLayer-4 | DecoderLayer-3 | $c_{in} = 256, c_{out} = 128$ | $[B, 128, 32, 32, 32]$ |
| Conv3D-1 | DecoderLayer-4 | kernel 1, stride 1, padding 0 | $[B, 1, 32, 32, 32]$ |

