# OpenReview forum: "POC-SLT: Partial Object Completion with SDF Latent Transformers"
_ICLR.cc/2025/Conference — Submitted to ICLR 2025_

### Official Review · Reviewer_jb2s · 2024-11-02

**Soundness:** 3
**Presentation:** 3
**Contribution:** 3
**Rating:** 6
**Confidence:** 3

**Summary:**

This paper introduces a framework for efficient completion of partial SDF. Its contributions are:
1. A novel framework for efficient completion of partial SDF using VAE which is described as Patch Variational Autoencoder (P-VAE) and MAE-like transformer architecture which is described as SDF-Latent-Transformer (SLT).
2. Evaluation of P-VAE and SLT on ShapeNet and ABC dataset that indicates that P-VAE is superior to some of the state-of-the-art methods.

This paper indicates the effectiveness of P-VAE and SLT with a lot of qualitative and quantitative results. But this method lacks some special designs for SDF and there are still some questions that need to be clarified.

**Strengths:**

1. This paper introduces VAE-like encoder-decoder architecture and MAE-like transformer architecture to efficiently complete partial SDF.
2. P-VAE and SLT have shown superior performance on ShapeNet and ABC dataset.

**Weaknesses:**

1. SDF represents the surface of an object with a signed distance function, and the value reflects the distance to the surface. However, P-VAE has no unique design for addressing the conflicts caused by inconsistent values in different SDF patches, which may harm the performance of shape completion based on SDF. The authors should discuss how to handle or mitigate potential inconsistencies between SDF patches or provide an analysis of how much this affects the completion results in practice.
2. In lines 250-251, the authors mention that position encoding is helpful for shape completion, but the paper lacks details like how to design the position embedding and some results that indicate it impacts shape completion. The authors should provide the exact formulation of the positional encoding and the ablation study showing the impact of including against excluding the positional encoding on completion quality.
3. In lines 306-307, measuring the completion performance by measuring how faithfully the geometry given in the input was preserved during the completion may not be enough. The metrics that evaluate the plausibility or realism of the completed regions should also be considered.
4. There are three types of completion, but there is no qualitative result about randomly removing the patches. The authors should include qualitative examples of the random patch removal completion task in their results section or supplementary materials.
5. The paper compares the quantitative results with some state-of-the-art methods only on chairs, which is insufficient to prove the effectiveness of P-VAE and SLT. More object categories should be considered.

**Questions:**

- See the weaknesses above.

---

> ### Author Response · Authors · 2024-11-22
>
> Dear Reviewer jb2s,
>
> thank you for reviewing our paper.
>
> In the following, we address your concerns and potential misconceptions about our method:
>
> # W1
> Concerning the input to our model,
> we demonstrated that the P-VAE can accurately encode highly detailed geometric information,
> providing the SLT with a rich representation for shape completion.
>
> Concerning the output, as you can see in our results (e.g., Figure 3),
> there are minor inconsistencies between the predicted patches, producing small seams.
> However, these seams are small enough that one can potentially remove them in a simple post-filtering process.
> The SLT naturally learns to predict patches that fit well together during training
> thanks to its global view in the transformer-encoder.
>
> To ensure seamless predictions without post-filtering,
> one can look into a convolutional decoder, accounting for neighboring patches,
> or interpolate between overlapping patches or multi-resolution patches.
> Given our almost seamless results, we did not see the need for applying such methods.
>
> # W2
> 3D positional encoding is necessary to spatially identify the individual patches in the sequence given to the transformer-encoder.
> The location of a patch within the sequence has no influence on the processing by the transformer-encoder.
> Therefore, without positional encoding, there is effectively no spatial relationship between patches.
> Crucially, all masked tokens would complete to the **exact same** result.
>
> We apply 3D positional encoding in a standard way, where the channels
> are equally split into representations of the $x,y,z$ coordinates, encoded as sine-waves of increasing frequency.
> The exact encoding is unlikely to matter too much in our case.
> It simply needs to describe the $4\times4\times4$ grid positions of the patches.
> In other, more common, applications of 3D positional encoding,
> one would typically represent varying 3D locations of arbitrary points in a point cloud.
>
> # W3
> We also prefer to report more meaningful metrics for shape completion,
> which is why we list a wide range of metrics when evaluating our method.
> However, previous methods did in fact not report these metrics on comparable tasks.
> Therefore, we had to fall back on reporting on whatever published data was available.
> The only compatible data turned out to be the UHD data reported by AutoSDF and SDFusion.
>
> We have since taken the time to run these evaluations for competing methods ourselves (Table 3, updated).
> Ideally, these metrics should have already been reported by the original authors
> to ensure that the best possible configuration of their method is used.
>
> # W4
> Results for shape completion from randomly removed patches are already shown in Figure 9 in the Appendix,
> matching the quantitative evaluation in Table 2, and also as part of the supplementary video.
>
> # W5
> Here, we again compare with all available published data.
> Previous methods only train and evaluate on very small subsets of data, containing mostly chairs,
> to keep their training and evaluation costs down.
> So, what is compared is the smallest common denominator.
> For our method, we also reported the result across all classes (previously Table 3, now Table 9).
>
> We have since taken the time to run the evaluation for AutoSDF and SDFusion
> to compare all metrics across all classes contained in their test set (Table 3, updated).
>
> Our method is the first to scale up to all of ShapeNet,
> so there is no published data to compare with, as we cannot possibly re-train other methods for comparison.
>
>
> We hope that this answers all of your concerns
> and allows you to wholeheartedly recommend our paper for acceptance.
>
> Best regards,
> the authors

---

> > ### Comment · Reviewer_jb2s · 2024-11-26
> > **Thanks.**
> >
> > Thanks for the detailed feedback from the authors. In the first review round, I raised 5 questions, mainly about the method and experimental analysis. Currently, the authors' feedback reasonably addresses my concerns. Thus, I still recommend a score of 6, leaning to accept this paper.

---

> > > ### Author Response · Authors · 2024-11-28
> > >
> > > Dear Reviewer jb2s,
> > >
> > > in your initial review, you pointed out several strengths about our paper.
> > > However, given your initial concerns, you decided to give a lower score.
> > > Since these concerns are now resolved, we would have expected a higher score.
> > >
> > > Is there any other concern that keeps you from raising your score?
> > > We would be happy to answer any of your questions.
> > >
> > > Best regards,
> > > the authors

---

### Official Review · Reviewer_1tgP · 2024-11-03

**Soundness:** 2
**Presentation:** 2
**Contribution:** 2
**Rating:** 3
**Confidence:** 3

**Summary:**

This paper introduces a shape completion method for high-resolution signed distance functions (SDFs), leveraging shared weights to efficiently represent small SDF patches.

The model encodes the entire shape as a sequence of latent codes generated by a patch-based variational autoencoder (P-VAE). Each patch is labeled as either "masked" (incomplete) or "non-masked" (complete), and shape completion is performed in latent space, with the result decoded back into an SDF using the P-VAE decoder. This patch-based approach allows for efficient inference and achieves high-quality results.

**Strengths:**

* Fast inference time: a key advantage over other sequential (eg autoregressive) approaches is utilizing the MAE decoder.
* Modular Patch-VAE architecture enables generalization by pre-training on small-scale patches, an effective component, even if previously used in related works.
* High-quality shape completion, especially in capturing fine-grained details.
* Simple yet effective approach, that avoids unnecessary complexity with a straightforward architecture and objective.

**Weaknesses:**

- Potential "leakage" issue: Non-masked voxels adjacent to masked patches may encode distances to missing parts, indirectly leaking information about regions to be completed. Discussing this limitation and potential remedies (e.g., using TSDF instead of SDF) would strengthen the work. Beyond conducting an ablation study of usage of SDF vs TSDF, first it should be qualitatively checked how much information is in fact encapsulated in non-masked patches, which regards the masked patches.
Masking in inference seems to require predefined regions marked as "known" or "unknown" potentially limiting the method. If so, this should be clarified as it diverges from traditional shape completion tasks, which typically address irregular partial shapes. You can also suggest possible modifications and future directions which allow general completion patterns.
- The approach is similar to pre-training on (T)SDF patches as done in [1]. This related work should be cited and discussed. Additionally, [1] demonstrated generalization to large scenes; it would be valuable to discuss whether the proposed method can achieve similar scalability.
- Limited model comparisons: The evaluation mainly compares against older methods like DeepSDF, omitting more recent approaches. To represent an implicit method such as DeepSDF, consider utilizing more recent architectures such as Neural Fields as Learnable Kernels for 3D Reconstruction.
- Comparison fairness: Competing models like SDFusion and PoinTR, unlike the proposed model, do not appear to receive explicit masking information on areas to be completed. If true, this gives the proposed model an advantage and should be addressed.
- Metrics: The evaluation relies heavily on UHD, which I believe is secondary in shape completion contexts, mainly measuring the fidelity of preserving the known region. Including additional metrics as is done in baselines would better reveal the model's performance. Those metrics can be all or at least some of the metrics you have used for your inner comparisons. Give precedence to important metrics regarding shape completion, such as Chamfer distance which is elementary as a main metric.
- The ordering of grid coordinates in the latent code (due to positional encoding in the transformer) suggests that the model expects canonically oriented shapes, which may be difficult to achieve in practice given partial input.

**Minor Points:**
- Please clarify the distinction between "transformer encoder" and "transformer decoder" and explain why the encoder allows single-step inference.

**Missing References:**
[1] _Local Implicit Grid Representations for 3D Scenes_
[2] _Neural Fields as Learnable Kernels for 3D Reconstruction_

**Questions:**

* Referring to the first weakness, it is hard to understand the function of masking in inference, perhaps should elaborate / clarify.
* Is using a VAE really necessary if the model is non-probabilistic?
* Is composing SDF from patch SDFs that simple? Intuitively it appears that this might cause issues near patch boundaries

---

> ### Author Response · Authors · 2024-11-23
>
> Dear Reviewer 1tgP,
>
> thank you for reviewing our paper.
>
> We have noticed some potential misunderstandings and would like to clarify them first
> and then address your concerns and questions below:
>
> # S1
> We do not use a "MAE decoder", only a MAE-style transformer-encoder.
> SDF patches are then decoded using the P-VAE's pre-trained decoder.
>
> # S2
> We are the first to train a VAE on 3D SDF patches.
> Other methods only used AE/AD, VQAE, VQDIF, but never a VAE.
> Unlike prior UNet-style 2D VAEs, we do not use skip-connections between encoder and decoder.
> Our VAE is free to allocate latent parameters across levels of detail, unrestricted by pre-defined partitions for residuals.
>
> LIG [1] explicitly shy away from training a VAE
> for 3D patch representations due to the difficulties involved in doing so.
> Our method overcomes these difficulties
> and we even evaluated autodecoding, which they similarly excluded.
>
> # W1
> The current design cannot avoid SDFs leaking information about neighboring cells.
>
> Our evaluation shows highly detailed completions far away from input patches.
> This indicates that our SLT's shape completion capabilities cannot possibly derive from potential leaks.
>
> Retraining our P-VAE and SLT for TSDFs would be costly.
> Truncation will necessarily negatively impact performance, since it removes information.
> However, one could not conclude that this would be due to leakage problems,
> since the SDF space constrains the P-VAE's latent space:
> Unlike for TSDFs, introducing surfaces to SDFs requires large changes.
> Similarly, a small deviation in TSDF latent codes might add unintended surfaces,
> complicating shape completion.
>
> # W2
> Our goal is to establish a backbone for shape completion.
> Solving real-world 3D scanning tasks requires further steps as already outlined in our Limitations and Conclusion sections.
>
> # W3
> Our P-VAE can easily scale to patches from entire scenes.
> Training the SLT to on significantly more latent codes
> might require a convolutional, or multi-scale, approach.
>
> We did initially filter out this method among others from the scene-level category.
> We now discuss this method in our related work.
>
> # W4
> Actually, we only compared against DeepSDF in the supplemental for SDF autoencoding (Table 7).
> We also compared shape autoencoding with the recent 3DShape2VecSet (Table 8).
>
> We also compared with more recent methods (AutoSDF, AnchorFormer).
> We have since added comparisons against ShapeFormer and compare more metrics with SDFusion,
> as recommended by other reviewers (Tables 3,4).
>
> # W5
> We compare against data published by SDFusion themselves on this particular task.
> They must have trained their model with this task in mind.
>
> Furthermore, partial inputs are still given within a normalized coordinate system
> where the model can easily figure out missing halves.
>
> By masking the entire half, our model also has to figure out where to complete the object.
>
> # W6
> We also prefer more meaningful metrics for shape completion.
> However, previous methods did in fact not report these metrics on comparable tasks.
> We have since run these evaluations for competing methods ourselves (Table 3, updated).
>
> # W7
> The canonical orientation is derived from ShapeNet.
> It was shown to improve reconstruction on objects from known classes.
> Like previous methods, which widely share this limitation, we make use of it for fair comparison.
> Where not given, the canonical orientation can be estimated by methods like 3DScan2CAD.
>
> # Q1
> Masking is a required part of the MAE pipeline.
> The transformer-encoder is always given a full sequence of patches.
> For every given input patch, an output patch is produced.
> During training, inputs are masked to train predictions of masked patches from the given patches.
> During inference, the same mechanism deterministically predicts the completed output.
> Without a masked patch as an input, there would be no output.
>
> # Q2
> The VAE is essential for producing a smooth latent space.
> This is essential for the attention mechanism at the core of the transformer architecture.
> With VAE, slightly different latent codes maintain the same meaning and receive similar attention.
> Without VAE, slightly different latent codes could have completely separate meanings, conflicting with the attention.
> Also, the smooth latent space generally helps with the training and relieves the transformer from having to predict exact latent codes.
>
> # Q3
> The SLT learns to predict patches that fit together almost perfectly (e.g., Figure 3), given enough training,
> despite never observing the decoded SDF values and potential differences between patch boundaries.
> We think that this is due to the rich information contained in the SDF patches (also in contrast to TSDFs),
> where transitions between patches are inherently smooth.
>
>
> We hope that this answers all your questions and concerns.
> We believe that your initial score is not appropriate for what we have demonstrably achieved with our method.
> Please reconsider it.
>
> Best regards,
> the authors

---

> > ### Author Response · Authors · 2024-11-28
> >
> > Dear Reviewer 1tgP,
> >
> > since we did not yet hear back from you,
> > we would like to ask if you have any further questions or concerns,
> > such that we can respond in time.
> >
> > Otherwise, we would be happy if you could increase your score
> > based on the answers we previously provided.
> >
> > Best regards,
> > the authors

---

### Official Review · Reviewer_mpJj · 2024-11-04

**Soundness:** 2
**Presentation:** 2
**Contribution:** 2
**Rating:** 5
**Confidence:** 5

**Summary:**

The paper works on the problem of partial object shape completion. Given a partial SDF volume, the proposed method first encodes the SDF volume into a latent code for each patch using the proposed Patch Variational Autoencoder(P-VAE), which encodes  SDF patches of resolution 32^3 into a latent code. 4^3 latent code encoded by the P-VAE can be used to train a masked autoencoder for the shape completion task. The proposed method achieved a significant reduction in the running speed thanks to its MAE architecture and short context length (64) for the transformer. The authors tested the proposed method on ShapeNet and ABC datasets and achieved better performance on metrics for reconstruction quality compared to previous methods. The qualitative study also demonstrated the results of reconstruction.

**Strengths:**

1. The authors proposed an efficient architecture that runs much faster than previous diffusion-based methods and auto-regressive-based methods and still demonstrated decent quantitative and qualitative results on shape completion benchmarks. The efficient design and fast running speed are appreciated for possible real-world applications.
2. This paper addressed an interesting problem of 3D shape completion, which could lead to possible applications in 3D reconstruction and robotics.
3. The authors compared the proposed method with the previous method quantitatively and qualitatively with high-quality renderings.

**Weaknesses:**

1. Lack of ablative study on the resolutions of the patch size and number of patches. The 32^3 SDF volume seems to be a relatively large SDF with lots of information. It will be good to have an ablative study with smaller patch sizes, such as 16^3 or 8^3.
2. The presentation of the proposed method is vague and confusing. It will be much easier for the reader to understand if the authors point out they only use a transformer with 8x8x8 context length, and each token encodes the information of a 32^3 SDF volume, and that's why the whole model is lightweight. It would be good if the pipeline figures and method section made this point clear.
3. Multiple publications have followed AutoSDF, such as DiffComplete, but the authors did not quantitatively compare to those newer publications, which makes it difficult for the reviewer to judge the technical improvements over that method.
4. Previous shape completion, like AutoSDF and Shapeformer, also demonstrated its power to generate multiple hypotheses when the partial observation is ambiguous. However, the proposed method did not discuss nor benchmark the task of multi-modal shape completion. Would it be possible to also evaluate total mutual difference(TMD)?
5. The authors provided autoencoding results in Table 4 but did not compare them against any baseline methods such as shapeformer, providing limited information on understanding the effectiveness of the power of presentation.

Minor issues:
- Please use the word autoencoder instead of "Auto Encoder"

**Questions:**

1. Can authors elaborate on the design choice for the patch size and size of the SDF volume and how changes to those hyper-parameters will affect the proposed method's performance?
2. The SDF volume contains much more information in the shape completion task than point cloud input because it hinted the location of the possible surface. How did the authors address this possible information leak in the evaluation benchmarks? Real-world sensors typically cannot provide SDF volume as an input because the location of the real surface is yet to be known; how could this method be applied to real-world 3D reconstruction applications?
3. Can authors provide the number of parameters in Table 1 to better understand differences compared to the previous method?
4. Can authors provide more information on the quantitative comparison to demonstrate the proposed method is comparable to more recent publications?

---

> ### Author Response · Authors · 2024-11-23
>
> Dear Reviewer mpJj,
>
> thank you for reviewing our paper. Your feedback is appreciated.
> We address your concerns below:
>
> # W1 / Q1
> The $32^3$ patch size represents a good level of detail for processing:
> The patch contains information about local geometry beyond just small surface patches, but not the global object.
>
> Using $16^3$ patches may still perform well,
> but $8^3$ patches likely provide too small-scale information to reason about effectively.
>
> Importantly, decreasing the patch size cubically increases the length of the input sequence given to the transformer.
> Using half-resolution patches might increase the cost by a power of $6$,
> since self-attention is an $O(N^2)$ operation.
> With our short sequence length, all patches are relevant,
> allowing for a cheap feed-forward transformer-encoder design using only self-attention between a small number of patches.
> Otherwise, more complicated methods might be needed, even requiring filtering the input
> and predicting potentially relevant output patches for querying a transformer-decoder.
>
> We aim to compare with a $16^3$ version in the camera-ready paper.
> Independent of this ablation study, we have demonstrated the effectiveness of our method
> and can only further validate our hyper-parameter choices.
>
> We did not increase the SDF resolution beyond $128^3$ to limit the storage and compute requirements for ground truth SDFs.
>
> # W2
> We rewrote the caption of Figure 1 and the method intro to make this more clear.
>
> # W3
> The problem definition of many recent works is incompatible with our method, which works on 3D patches.
> Comparing with their published numbers would be problematic due to the differences in the evaluated tasks.
> We can also not exchange inputs between models, since they operate on different resolutions
> and have not been trained for each other's tasks and type of data.
>
> In particular, DiffComplete train on 3D-EPN, containing partial scans,
> while we only consider complete patches.
> Also, we operate on $128^3$ SDFs, while they work with $32^3$ TSDFs and TUDFs.
> In order to meaningfully process their inputs, we would first have to regenerate them at higher resolution.
>
> For more recent comparisons,
> we already compared against AnchorFormer in Figure 3 and Table 10 (previously Table 9).
> and now also evaluated AutoSDF and SDFusion for all metrics to provide the metrics missing in their publications (Table 3, updated).
> Additionally, we compare our P-VAE against the VQDIF from ShapeFormer in Table 4 (updated, previously Table 3) and Figure 6 (updated).
>
> Our goal is to deterministically generate the best possible completion of the given input, not multi-modal shape generation.
> Therefore, evaluating TMD would not be meaningful.
>
> # W4
> See Table 4 (updated) and Figure 6 (updated) for a comparison against the VQDIF by ShapeFormer.
> In contrast to our method, which processes a full $128^3$ SDF per object,
> their method only takes 32k surface points per object as input, limiting the expressiveness of the their model.
>
> We did already compare our P-VAE's quality
> against DeepSDF and 3DShape2VecSet as baselines in Tables 7 and 8 in Appendix A.4.
>
> # Q2
> The current design cannot avoid SDFs leaking information about neighboring cells.
>
> We show evaluations given sparse random inputs or single octants,
> where the completed volume is significantly larger than the volume affected by information leaks.
> Even when given the bottom half, none of the detail in the top-half completion could be inferred by these leaks.
> Based on the high amount of detail we observed in the results far from the input,
> we did not consider this worthy of further investigation.
>
> # Q3
> Even from partial observations, one can compute partial meshes, from which partial SDFs can be computed.
> These will contain some artifacts due to missing information.
> If one has sufficiently complete information about several $32^3$ patches,
> one can still feed them into the SLT, while masking out all other patches to perform shape completion.
> E.g., if one has particularly clean information about one side of the object, but did not fully observe the other half.
>
> If the scan is too incomplete to extract any complete patches,
> one could try to estimate matching complete patches from partial inputs, e.g., using the approach presented by DeepSDF.
> Alternatively, one could try to retrain or fine-tune the SLT on partially corrupted patches.
>
> # Q4
> We have updated Table 1 to include this information.
>
> # Q5
> We ran evaluations for AutoSDF and SDFusion to produce the missing metrics on the shape completion task
> compared in Table 3 (updated).
>
> In general, running these comparisons is problematic due to differences in problem definitions,
> data preprocessing, train/test splits and evaluation metrics and scales.
> A well-defined set of standard benchmark tasks is desperately needed to allow for meaningful evaluations in this field.
>
>
> We hope this addresses all of your questions and clears any concerns that resulted in a lower score.
>
> Best regards,
> the authors

---

> > ### Comment · Reviewer_mpJj · 2024-11-26
> > **Thanks for the reply; Unclear about the input of partial SDF**
> >
> > The reviewer would like to thank the authors for the clarifications. The adjustment of tables and figures makes this work more comprehensible. However, the reviewer is still confused about the evaluation protocols with random input and single octant. Did the authors compute the partial mesh from partial observation to generate the SDF volume as the method's input? It's unclear how the model will complete a shape if the SDF is given from partial mesh. For example, a point in the center of an object will have a relatively large SDF value from the complete SDF volume but near zero SDF from the partially completed mesh. How will the model deal with this case if it's not explicitly trained? The reviewer will consider raising the rating if this concern is also addressed.

---

> > > ### Author Response · Authors · 2024-11-27
> > >
> > > Dear Reviewer mpJj,
> > >
> > > the input to our model is always a combination of ground truth SDF patches and mask tokens, in line with the MAE design.
> > >
> > > While it would be possible to compute SDF patches from partial meshes to some extent,
> > > we always use patches extracted from the full mesh and then apply masking to represent a partial mesh.
> > > This applies in the same way to all settings (bottom half, octant, random).
> > >
> > > It would be impractical to have to generate SDFs from partial meshes during training, due to the cost involved.
> > > With this design, we only need one set of pre-computed latent SDF patches per mesh
> > > in order to train the SLT on arbitrarily masked input settings.
> > >
> > > If one wanted to compute SDF patches from partial meshes,
> > > only those patches representing complete information
> > > would be directly applicable to the SLT given its current training regime.
> > > To produce more useful patches from a partial observation,
> > > one might be able to use the DeepSDF method to regress to complete SDF patches given partial information
> > > or fine-tune the SLT on partial inputs, as outlined in our answer to question 3.
> > >
> > > Please feel free to reach out if you have any more questions.
> > >
> > > Best regards,
> > > the authors

---

> ### Comment · Reviewer_mpJj · 2024-11-30
>
> Dear Authors,
>
> Thank you for your explanation, which cleared up my doubts. While it makes sense to use the SDF patch from the full mesh during training time, I am afraid it's an unfair comparison with the point cloud based method in empirical experiments during the test time. The author is concerned that this method is not directly applicable to real-world vision and robotics tasks as it assumes an SDF volume from a full mesh. Due to this limitation, the author is not convinced of the experimental advances in this quantitative comparison and will keep the rating.
>
> Best,
> Reviewer

---

> > ### Author Response · Authors · 2024-12-02
> >
> > Dear Reviewer mpJj,
> >
> > we primarily compare our method against AutoSDF and SDFusion (in Table 3),
> > which are trained and evaluated in exactly the same setting:
> > SDF patches are computed from complete meshes and then masked for completion.
> > We followed their task design for a fair comparison.
> >
> > We only included visual results for a recent point cloud method (AnchorFormer)
> > in the main paper for completeness.
> > We mention details about that comparison in Appendix 6,
> > where we explicitly state that this is somewhat a problematic comparison
> > due to the different modalities and tasks.
> > We made our best efforts for a fair comparison
> > and do not think it is valid to say that any of our comparisons are unfair.
> >
> > While not yet directly applicable to real-world tasks,
> > our method can be an important stepping stone towards effective shape completion.
> > We have shown that our latent SDF space can represent highly detailed objects
> > and can be processed efficiently with the SLT design.
> > With additional pre-processing, it can be retooled for real-world applications in future work.
> >
> > Best regards,
> > the authors

---

### Official Review · Reviewer_AoP9 · 2024-11-05

**Soundness:** 2
**Presentation:** 3
**Contribution:** 1
**Rating:** 5
**Confidence:** 5

**Summary:**

This paper proposes a sdf latent transformer for partial object completion. Specifically, it first divides high-resolution voxels into multiple smaller voxels and learns latent representations $z$ of these smaller voxels using a vae. Then, these latent representations are arranged as a sequence and processed by a masked transformer, which shares similarities with masked autoencoder. The experiments are conducted on three datasets for the completion task. The experimental results are better than those of the compared approaches.

**Strengths:**

1. The paper is easy to understand.
2. The experimental results are better than the compared methods.

**Weaknesses:**

The novelty of this paper is very limited.

1. The division of high-resolution voxels into smaller patches is reasonable but is straightforward and thus of limited innovation.

2. The p-VAE is almost identical to the original vae without any adaptation or improvements for this task.

3. The SDF-Latent-Transformer idea is very similar to the masked autoencoder [1], so it lacks novelty.

In short, the method proposed in this paper is somewhat like a combination of different well-known models, therefore, the novelty is the main concern.

[1] He K, Chen X, Xie S, Li Y, Dollár P, Girshick R. Masked autoencoders are scalable vision learners. InProceedings of the IEEE/CVF conference on computer vision and pattern recognition 2022 (pp. 16000-16009).

**Questions:**

There is no questions, as the method proposed in this paper combines different existing models and therefore straightforward.

---

> ### Author Response · Authors · 2024-11-22
>
> Dear Reviewer AoP9,
>
> thank you for taking the time to review our paper.
>
> From your review, we understand that you would have liked to see advancements in the underlying machine learning methodology.
> Our focus lies on the application side.
> Based on the call for papers, which lists "applications in vision" and "representation learning for computer vision",
> this should be considered a relevant subject area.
>
> To establish the universal SDF-patch latent space that our SLT operates on,
> our P-VAE was trained on millions of randomly cropped 3D patches from ShapeNet,
> which we deliberately selected from various anisotropic scales and with random offsets to incorporate all topological settings.
> As a result, our latent space allows for autoencoding practically arbitrary 3D patches with very little loss in quality,
> even when containing intricate geometry.
> It can be meaningfully processed downstream, e.g., by the SLT,
> and has shown to generalize well towards ABC and even Objaverse.
>
> Our P-VAE has been custom-designed to contain skip connections only within the encoder and decoder, but not across them
> to compress the large 3D volume down to a small latent code.
> Using a standard UNet-style VAE like in the 2D image domain would primarily learn residuals
> and not compress the input down into a single representation.
> This allows the P-VAE to freely allocate latent parameters to the necessary levels of detail.
>
> We are the first method to successfully train a VAE on 3D SDF patches.
> Previous methods like "Local Implicit Grid Representations for 3D Scenes" explicitly shy away from this,
> citing "DeepSDF" as "[observing] the stochastic nature of the VAE [making] training difficult".
>
> Our results demonstrate that our P-VAE achieves significantly better autoencoding quality than previous work.
> We already demonstrated this in comparison with 3DShape2VecSet (Table 8) and DeepSDF (Table 7),
> and now further validated that against ShapeFormer (Table 4, updated).
>
> In addition to the high-quality autoencoding,
> our latent codes are also well-suited for 3D shape completion, as shown by the compelling results of our SLT,
> which even manages to produce almost seamless results without complicating the architecture with ad-hoc filtering modules.
>
> In general, we think that conceptually simple, yet effective methods like ours
> should be preferred to more complicated methods.
> Given that our method still outperforms prior work without further tweaking
> is a result of our deliberate design choices.
> It provides a solid baseline for further research down this line
> and thus represents meaningful progress towards better 3D shape completion.
>
> Working with SDFs is also technically challenging, since they are costly to compute
> and require large amounts of storage and high bandwidth for training.
> Existing software computing exact SDFs is prohibitively slow
> and would have not allowed us to process all of ShapeNet and 100k meshes from ABC.
> Competing (T)SDF-based methods still mostly operate at only $32^3$ input resolution (some at $64^3$)
> or on small subsets of ShapeNet, because they shy away from
> computing and processing SDFs at high resolutions from high-resolution meshes at scale.
> E.g., several current methods (such as DiffComplete) still rely on the now 7 years old pre-computed $32^3$ (T)SDFs from the 3D-EPN dataset.
>
> For training the SLT, we had to compute the $128^3$ ground truth SDFs for all 150k meshes.
> For training the P-VAE, we had to compute $32^3$ ground truth SDFs for millions of sampled patches.
> To facilitate this, we first implemented our own parallelized C++ code,
> which we have since further replaced with our own CUDA code.
> Both heavily utilize spatial acceleration structures to speed up the computations.
> Otherwise, this would have not been possible at this scale.
>
>
> If seen narrowly enough, every new method is just a combination of old methods.
> This is how we, as the research community, make progress.
> We hope that you reconsider your decision.
>
> Best regards,
> the authors

---

> ### Comment · Reviewer_AoP9 · 2024-11-26
>
> Dear Authors,
>
> I appreciate your response. After reading the response, let me summarize two key points: 1. the proposed model is trained on millions of patches from 3D shapes, which is at a larger scale than previous methods; 2. the proposed model can achieve high-resolution SDFs, 128^3 rather than 32^3 in previous methods. Therefore, the proposed method can achieve superior performance on downstream tasks, such as shape completion.
>
> Indeed, I did notice these two key points during the first round of review. However, I want to see some technical (or methodology) novelty, not just a large pre-trained 3D vision model. I acknowledged the value of the high-resolution SDF model as a strong baseline as mentioned in the response, therefore, I raised my score to 5.

---

### Meta-Review · Area_Chair_DZ4V · 2024-12-14

**Metareview:**

The paper describes an autoencoder for 3D shape patches that allows to train a masked autoencoder-like transformer for shape completion.
The reviews of this work are negatively leaning, being concerned about technical novelty, missing applicability to real data and insufficient comparisons.
After taking a closer look at paper and discussions, I agree with the general sentiment of the reviewers and would follow with a reject recommendation.

**Additional Comments On Reviewer Discussion:**

Pre-rebuttal, the reviewers had concerns regarding technical novelty and missing comparisons / comparison metrics. During the rebuttal, some of these concerns were addressed by adding additional comparisons against baselines and additional evaluated metrics. Also several questions about unclear details have been resolved. However, all in all, the authors did not manage to convince the reviewers to change their assessment to an accept recommendation.

---

### Decision · Program_Chairs · 2025-01-22

Reject